# Learning Good Interventions in Causal Contextual Bandits with Adaptive Context

## Abstract

We study a variant of causal contextual bandits where the context is stochastically dependent on an initial action chosen by the learner. This adaptive context setting allows the environment to elicit some initial choice from the learner before providing the context. Upon observing the context, the learner picks another action (an intervention in a causal graph) based on which they receive a reward. The objective is to identify near-optimal atomic causal interventions at the initial state and post context identification, to maximize reward. We extend prior work from the deterministic context setting to obtain simple regret minimization guarantees. This is achieved through an instance-dependent causal parameter, $\lambda$, which characterizes our upper bound. Furthermore, we prove that our simple regret is essentially tight for a large class of instances. A key feature of our work is that we use convex optimization to address the bandit exploration problem. We also conduct experiments to validate our theoretical results

## 1 Introduction

Recent years have seen an active interest in causal bandits from the research community (Lattimore et al., 2016; Sen et al., 2017a;b; Lee & Bareinboim, 2018; Yabe et al., 2018; Lee & Bareinboim, 2019; Lu et al., 2020; Nair et al., 2021; Lu et al., 2021; 2022; Maiti et al., 2022; Varici et al., 2022; Subramanian & Ravindran, 2022; Xiong & Chen, 2023). In this setting, one assumes an environment comprising of causal variables that are random variables that influence each other as per a given causal (directed, and acyclic) graph. Specifically, the edges in the causal DAG represent causal relationships between variables in the environment. If one of these variables is designated as a reward variable, then the goal of a learner then is to maximize their reward by *intervening* on certain variables (i.e., by fixing the values of certain variables). The rest of the variables, that are not intervened upon, take values as per their conditional distributions, given their parents in the causal graph. In this work, as is common in literature, we assume that the variables take values in $\{0, 1\}$. Of particular interest are causal settings wherein the learner is allowed to perform *atomic interventions*. Here, at most one causal variable can be set to a particular value, while other variables take values in accordance with their underlying distributions.

It is relevant to note that when a learner performs an intervention in a causal graph, they get to observe the values of multiple other variables in the causal graph. Hence, the collective dependence of the reward on the variables is observed through each intervention. That is, from such an observation, the learner may be able to make inferences about the (expected) reward under other values for the causal variables (Peters et al., 2017). In essence, with a single intervention, the learner is allowed to intervene on a variable (in the causal graph), allowed to observe all other variables, and further, is privy to the effects of such an intervention. Indeed, such an observation in a causal graph is richer than a usual sample from a stochastic process. Hence, a standard goal in causal bandits is to understand the power and limitations of interventions. This goal manifests in the form of developing algorithms that identify intervention(s) that lead to high rewards, while using as few observations/interventions as possible. We use the term *intervention complexity* (rather than sample complexity) for our algorithm, to emphasize that interventions are richer than samples.

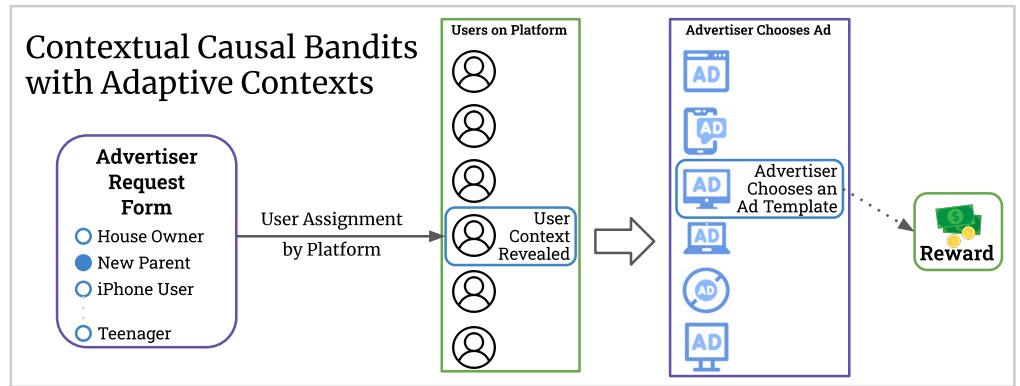

Figure 1: Flowchart illustrating the decision-making process of an advertiser posting ads on a platform like Amazon, and the subsequent interaction with the platform.

In the learning literature, there are several objectives that an algorithm designer might consider. Cumulative regret, simple regret, and average regret have prominently been studied in literature (Lattimore & Szepesvári, 2020; Slivkins et al., 2019). In this work we focus on minimizing simple regret, wherein the algorithm is given a time budget, up to which it may explore, at which time it has to output a near-optimal policy.

Addressing causal bandits, the notable work of Lattimore et al. (2016) obtains an intervention-complexity bound for minimizing simple regret with a focus on atomic interventions and parallel causal graphs. Maiti et al. (2022) extend this work to obtain intervention-complexity bounds for simple regret in causal graphs with unobserved variables. The work by Lu et al. (2022) extends this setting to causal Markov decision processes (MDPs), while addressing the cumulative regret objective. Combinatorial causal bandits have been studied by Feng & Chen (2023) and Xiong & Chen (2023).

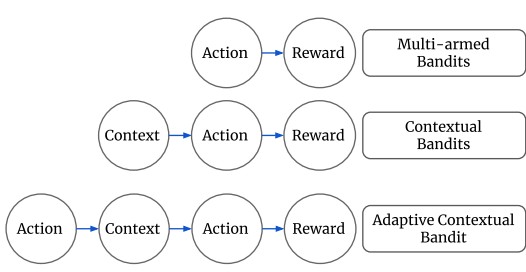

Figure 2: In contextual bandits with adaptive contexts, the environment provides a context based on an initial action chosen by the learner. In *causal* contextual bandits, such actions are interventions in causal graphs.

Causal contextual bandits have been studied by Subramanian & Ravindran (2022) where the contexts may be chosen by the learner (rather than be provided by the environment). Here we generalize Subramanian & Ravindran (2022) to a setting where the context is provided by the environment, adaptively, in response to an initial choice of the learner.

**Motivating Example:** Consider an advertiser looking to post ads on a web-page, say Amazon. They may make requests for a certain type of user demographic to Amazon. Based on this initial request, the platform may actually choose one particular user to show the ad to. At this time, certain details about the user are revealed to the advertiser. For example, the platform may reveal some of the user demographics, as well as certain details about their device. Based on these details, the advertiser may choose one particular ad to show the user. In case the user clicks the ad, the advertiser receives a reward. The goal of the learner is to find optimal choices for initial user preference, as well as ad-content such that user clicks are maximized. We illustrate this example through Figure 1 where we indicate the choices available for template and content interventions.

## 1.1 Our Contributions

We develop an algorithm to identify near-optimal interventions in causal bandits with adaptive context, and show that the simple regret of such an algorithm is indeed tight for several instances. We highlight the main contributions of our work below.

**1.** We develop and analyze an algorithm for minimizing simple regret for causal bandits with adaptive context in an intervention efficient manner. We provide an upper-bound on intervention complexity in Theorem 1.

**2.** Interestingly, the intervention complexity of our algorithm depends on an instance dependent structural parameter—referred to as $\lambda$ (see equation (3))— which may be much lower than $nk$, where $n$ is the number of interventions and $k$ is the number of contexts.

**3.** Notably, our algorithm uses a convex program to identify optimal interventions. Unlike prior work that uses optimization to design exploration (for example see Yabe et al. (2018)), we show (in Appendix Section E) that the optimization problem we design is convex, and is thus computationally efficient. Using convex optimization to design efficient exploration is in fact a distinguishing feature of our work.

**4.** We provide lower bound guarantees showing that our regret guarantee is tight (up to a log factor) for a large family of instances (see Section 4 and Appendix Section F).

**5.** We demonstrate using experiments (see Section 5) that our algorithm performs exceeding well as compared to other baselines. We note that this is because $\lambda \ll nk$ for $n$ causal variables and $k$ contexts.

In conclusion, we provide a novel convex-optimization based algorithm for Causal MDP exploration. We analyze the algorithm to come up with an instance dependent parameter $\lambda$. Further, we prove that our algorithm is sample efficient (see Theorems 1 and 2).

## 1.2 ADDITIONAL RELATED WORK

| Description | Reference |
|---|---|
| Simple regret for bandits with parallel causal graphs | Lattimore et al. (2016) |
| Simple regret for atomic soft interventions | Sen et al. (2017a) |
| Simple regret for non-atomic interventions in causal bandits | Yabe et al. (2018) |
| Cumulative regret for general causal graphs | Lu et al. (2020) |
| Simple regret in the presence of unobserved confounders | Maiti et al. (2022) |
| Cumulative regret for unknown causal graph structure | Lu et al. (2021) |
| Cumulative regret for causal contextual bandits with latent confounders | Sen et al. (2017b) |
| Simple and cumulative regret for budgeted causal bandits | Nair et al. (2021) |
| Cumulative regret for Linear SEMs | Varici et al. (2022) |
| Best-intervention for combinatorial causal bandits | Xiong & Chen (2023) |
| Cumulative regret for combinatorial causal bandits | Feng & Chen (2023) |
| Cumulative regret for Causal MDPs | Lu et al. (2022) |
| Simple regret for causal contextual bandits | Subramanian & Ravindran (2022) |
| Simple regret for causal contextual bandits with adaptive context | **Our work** |

Table 1: Summary of prior work in causal bandits

Ever since the introduction of the causal bandit framework by Lattimore et al. (2016), we have seen multiple works address causal bandits in various degrees of generality and using different modelling assumptions. Sen et al. (2017a) addressed the issue of soft atomic interventions using an importance sampling based approach. Soft interventions in the linear structural equation model (SEM) setting was addressed recently by Varici et al. (2022). Yabe et al. (2018) proposed an optimization based approach for non-atomic interventions. This work was extended by Xiong & Chen (2023) to provide instance dependent regret bounds. They also provide guarantees for binary generalized linear models (BGLMs). The question of unknown causal graph structure was addressed by Lu et al. (2021), whereas Nair et al. (2021) study the case where interventions are more expensive than observations.

Maiti et al. (2022) addressed simple regret for graphs containing hidden confounding causal variables, while cumulative regret in general causal graphs was addressed by Lu et al. (2020). A notable work by Lu et al. (2022) formulates the framework for causal MDPs, and they

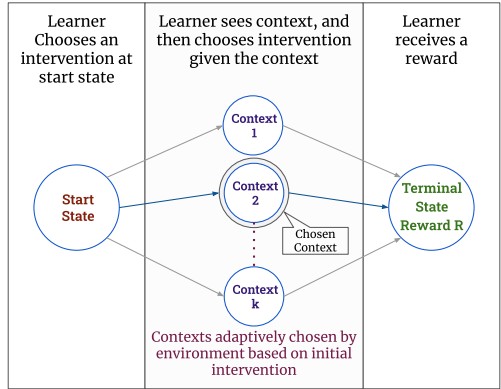 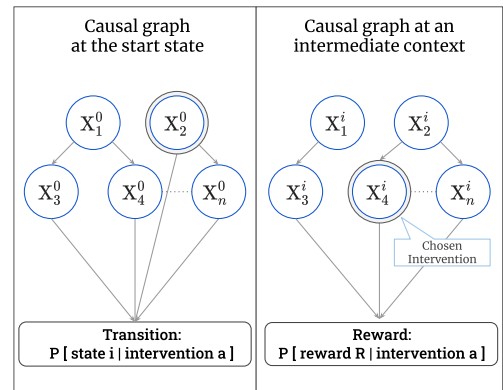

(a) Illustrative figure for causal contextual bandit with adaptive context.

(b) Illustrative Figure for Causal Graph at start state and at some intermediate context $i \in [k]$.

Figure 3: The transition to a particular context (chosen context in the figure on the left) is decided by the environment, whereas the interventions at the start state and an intermediate context (chosen interventions in the figure on the right) are chosen by the learner.

provide cumulative regret guarantees in this setting. Causal contextual bandits were addressed by Subramanian & Ravindran (2022); Sen et al. (2017b), and we extend these works to adaptive contexts.

We summarize the main works in this thread in Table 1 and provide a more detailed set of related work in Appendix A.

## 2 NOTATIONS AND PRELIMINARIES

We model the causal contextual bandit with adaptive context as a contextual bandit problem with a causal graph corresponding to each context. The actions at each context are given by interventions on the causal graph. Additionally, we have a causal graph at the start state, and the context is stochastically dependent on the intervention on the causal graph at the start state. For ease of notation, we will call the start state of the learner as context 0. The agent starts at context 0, chooses an intervention, then transitions to one of $k$ contexts $[k] = \{1, \ldots, k\}$, chooses another intervention, and then receives a reward; see Figure 3(a).

**Assumptions on the Causal Graph:** Formally, let $\mathcal{C}$ be the set of contexts $\{0, 1, \ldots, k\}$. Then, at each context, there is a Causal Bayesian Network (CBN) represented by a causal graph; see Figure 3(b). In particular, at each context $i \in \mathcal{C}$, the causal graph is composed of $n$ variables $\{X_1^i, \ldots, X_n^i\}$. For each $X_j^i \in \{0, 1\}$, with an associated conditional probability. We make the following mild assumptions on the causal graph at each context.

1. The distribution of any node $X_i$ conditioned on it's parents in the causal graph is a Bernoulli random variable with a fixed parameter.

2. The causal graph at each context is semi-Markovian. This is equivalent to making the following assumptions on the graph. No hidden variable in the graph has a parent. Further, every hidden variable has at most two children, both observable.

3. We transform the causal graph for each context as follows: For every hidden variable with two children, we introduce bidirected edges between them. If no path of bidirected edges exists between an intervenable node and its child, the graph is identifiable – a necessary and sufficient condition for estimating the graph's associated distribution.(Tian & Pearl, 2002).

**Interventions:** Furthermore, we are allowed atomic interventions, i.e., we can select *at most* one variable and set it to either 0 or 1. We will use $\mathcal{A}_i$ to denote the set of atomic interventions available at context $i \in \{0, \ldots, k\}$; in particular, $\mathcal{A}_i = \{do()\} \cup \{do(X_j^i = 0), do(X_j^i = 1)\}$ for $j \in [n]$. We note that $do()$ is an empty intervention that

Table 2: Summary of notations for our paper

| Notation | Explanation |
|---|---|
| Context 0 | Start state |
| Context $[k]$ | Intermediate contexts $\{1, \ldots, k\}$ |
| $X_j^i$ | Causal Variables: $X_j^i \in \{0, 1\}$  for all $i \in [k]$, $j \in [n]$ |
| $do(\cdot)$ | An atomic intervention of the form $do()$, $do(X_j^i = 0)$ or $do(X_j^i = 1)$ |
| $\mathcal{A}_i$ | Set of atomic interventions at context $i$ |
| $N$ | $N := |\mathcal{A}_i| = 2n + 1$  for all $i \in [k]$ |
| $R_i$ | Reward on transition from context $i$ |
| $m_i$ | Causal observational threshold at context $i \in \{0, \ldots, k\}$ |
| $M$ | diagonal matrix of $m_i$ values |
| $P \in \mathbb{R}^{N \times k}$ | Transition probabilities matrix: $\left[ P_{(a,i)} = \mathbb{P}\{i \mid a\} \right]_{a \in \mathcal{A}_0, i \in [k]}$ |
| $p_+$ | Transition threshold $p_+ = \min\{P_{(a,i)} \mid P_{(a,i)} > 0\}$ |
| $\pi : \mathcal{C} \to \mathcal{A}$ | Policy, a map from contexts to interventions. i.e. $\pi(i) \in \mathcal{A}_i$ for $i \in \{0\} \cup [k]$ |
| $\mathbb{E}\left[R_i \mid \pi(i)\right]$ | Expectation of the reward at context $i$ given intervention $\pi(i)$ |

allows all the variables to take values from their underlying conditional distributions. Also, $do(X_j^i = 0)$ and $do(X_j^i = 1)$ set the value of variable $X_j^i$ to 0 and 1, respectively, while leaving all the other variables to independently draw values from their respective distributions. Note that for all $i \in [k]$, we have $|\mathcal{A}_i| = 2n + 1$. Write $N := 2n + 1$.

**Reward:** The environment provides the learner with a $\{0, 1\}$ reward upon choosing an intervention at context $i \in [k]$, which we denote as $R_i$. Note that $R_i$ is a stochastic function of variables $X_1^i, \ldots, X_n^i$. In particular, for all $j \in [n]$ and each realization $X_j^i = x_j \in \{0, 1\}$, the reward $R_i$ is distributed as $\mathbb{P}\{R_i = 1 \mid X_1^i = x_1, \ldots, X_n^i = x_n\}$.

Given such conditional probabilities, we will write $\mathbb{E}[R_i \mid a]$ to denote the expected value of reward $R_i$ when intervention $a \in \mathcal{A}_i$ is performed at context $i \in [k]$. Here the expectation is over the parents of the variable $R_i$ in the causal graph, with the intervened variable set at the required value. Note that these parents (of $R_i$) may in turn have conditional distributions given their parents. The leaf nodes of the causal graph are considered to have unconditional Bernoulli distributions. For instance, $\mathbb{E}[R_i \mid do(X_j^i = 1)]$ is the expected reward when variable $X_j^i$ is set to 1, and all the other variables independently draw values from their respective (conditional) distributions. Indeed, the goal of this work is to develop an algorithm that maximizes the expected reward at context 0.

**Causal Observational Threshold:** We denote by $m_i$, the causal observational threshold[1] from Maiti et al. (2022) at context $i$. The existence of such a threshold at each context is guaranteed by the assumptions we made on the CBNs. In addition, let $M \in \mathbb{N}^{k \times k}$ denote the diagonal matrix of $m_1, \ldots, m_k$.

**Transitions at Context 0:** At context 0, the transition to the intermediate contexts $[k]$ stochastically depends on the random variables $\{X_1^0, \ldots, X_n^0\}$. Here, $\mathbb{P}\{i \mid a\}$ denotes the probability of transitioning into context $i \in [k]$ with atomic intervention $a \in \mathcal{A}_0$; recall that $\mathcal{A}_0$ includes the do-nothing intervention. We will collectively denote these transition probabilities as matrix $P := \left[ P_{(a,i)} = \mathbb{P}\{i \mid a\} \right]_{a \in \mathcal{A}_0, i \in [k]}$. Furthermore, write the transition threshold $p_+$ to denote the minimum non-zero value in $P$. Note that matrix $P \in \mathbb{R}^{|\mathcal{A}_0| \times k}$ is fixed, but unknown.

**Policy:** A map $\pi : \{0, \ldots, k\} \to \mathcal{A}$, between contexts and interventions (performed by the algorithm), will be referred to as a policy. Specifically, $\pi(i) \in \mathcal{A}_i$ is the intervention at context $i \in \{0, 1, \ldots, k\}$. Note that, for any policy $\pi$, the expected reward, which we denote as $\mu(\pi)$, is equal to $\sum_{i=1}^{k} \mathbb{E}\left[R_i \mid \pi(i)\right] \cdot \mathbb{P}\{i \mid \pi(0)\}$. Maximizing expected reward, at each intermediate context $i \in [k]$, we obtain the overall optimal policy $\pi^*$ as follows. For $i \in [k]$:

---

[1] Maiti et al. (2022) extend the causal observational threshold from Lattimore et al. (2016) to the general setting of causal graphs with unobserved confounders

$$\pi^*(i) = \underset{a \in \mathcal{A}_i}{\arg\max} \ \mathbb{E}\left[R_i \mid a\right] \tag{1}$$

$$\pi^*(0) = \underset{b \in \mathcal{A}_0}{\arg\max}\left(\sum_{i=1}^{k} \mathbb{E}\left[R_i \mid \pi^*(i)\right] \cdot \mathbb{P}\{i \mid b\}\right) \tag{2}$$

Our goal then is to find a policy $\pi$ with (expected) reward as close to that of $\pi^*$ as possible.

**Simple Regret:** Conforming to the standard *simple-regret* framework, the algorithm is given a time budget $T$, i.e., the learner can go through the following process $T$ times — (a) start at context 0. (b) Choose an intervention $a \in \mathcal{A}_0$. (c) Transition to context $i \in [k]$. (d) Choose an intervention $a \in \mathcal{A}_i$. (e) Receive reward $R_i$. At the end of these $T$ steps, the goal of the learner is to compute a policy. Let the policy returned by the learner be $\widehat{\pi}$. Then the simple regret is defined as the expected value: $\mathbb{E}[\mu(\pi^*) - \mu(\widehat{\pi}]$. Our algorithm seeks to minimize such a simple regret.

## 3  MAIN ALGORITHM AND ITS ANALYSIS

We now provide the details relating to our main Algorithm, viz. CONVEXPLORE. This Algorithm relies on three helper Algorithms which are detailed in Section B of the Appendix.

---

**Algorithm 1** CONVEXPLORE: Convex Exploration Algorithm

---

1: **Input:** Total rounds $T$
2: Perform actions at context 0 in a round robin manner for time $\frac{T}{3}$, and **estimate the transition probabilities** $P$ to the intermediate contexts on performing each intervention. Call this estimate $\widehat{P}$
3: Compute the frequency of performing each action, that maximizes the minimum frequency of visitation across contexts. Let $\tilde{f} \leftarrow \underset{\text{fq. vector } f}{\arg\max} \ \underset{\text{contexts } [k]}{\min} \ \widehat{P}^\top f$
4: Perform actions at context 0 with frequency $\tilde{f}$ for time $\frac{T}{3}$, and **estimate the causal observational threshold** matrix $M$ for the intermediate contexts, where $M$ is diagonal.
5: Compute the frequency of performing each action, that maximizes the minimum frequency of observation of actions at the intermediate contexts.
   Let $\widehat{f}^* \leftarrow \underset{\text{fq. vector } f}{\arg\min} \ \underset{\text{interventions } \mathcal{I}_0}{\max} \ \widehat{P}\hat{M}^{1/2}\left(\widehat{P}^\top f\right)^{\circ - \frac{1}{2}}$. This computation is efficient as the the program is a Convex Program.
6: Perform actions at context 0 with frequency $\widehat{f}^*$ for time $\frac{T}{3}$, and then **estimate the reward** matrix $\widehat{\mathcal{R}}$ for actions at the intermediate contexts.
7: **Estimate the optimal action at each intermediate context** $\widehat{\pi}(i) \forall i \in [k]$ based on $\widehat{\mathcal{R}}$. Let the estimate of optimal reward be $\widehat{\mathcal{R}}(\widehat{\pi}(i))$.
8: **Estimate the optimal action at the start context** $\widehat{\pi}(0)$, based on the transition probabilities $\widehat{P}$ and the optimal reward estimates $\widehat{\mathcal{R}}(\widehat{\pi}(i))$.
9: **return** $\widehat{\pi} = \{\widehat{\pi}(0), \widehat{\pi}(1), \dots, \widehat{\pi}(k)\}$ .

---

[a] We show detailed Algorithms for estimation of transition probabilities $P$ (line 2), estimation of causal observational threshold $M$ (line 4), and estimation of rewards $\mathcal{R}$ (line 6) in Appendix B

Our algorithm (CONVEXPLORE) uses subroutines to estimate the transition probabilities, the causal parameters, and the rewards. From these, it outputs the best available interventions as its policy $\widehat{\pi}$. Given time budget $T$, the algorithm uses the first $T/3$ rounds to estimate the transition probabilities (i.e., the matrix $P$) in Algorithm 2. The subsequent $T/3$ rounds are utilized in Algorithm 3 to estimate causal parameters $m_i$s. Finally, the remaining budget is used in Algorithm 4 to estimate the intervention-dependent reward $R_i$s, for all intermediate contexts $i \in [k]$.

To judiciously explore the interventions at context 0, CONVEXPLORE computes frequency vectors $f \in \mathbb{R}^{|\mathcal{A}_0|}$. In such vectors, the $a$th component $f_a \geq 0$ denotes the fraction of

time that each intervention $a \in \mathcal{A}_0$ is performed by the algorithm, i.e., given time budget $T'$, the intervention $a$ will be performed $f_a T'$ times. Note that, by definition, $\sum_a f_a = 1$ and the frequency vectors are computed by solving convex programs over the estimates. The algorithm and its subroutines throughout consider empirical estimates, i.e., find the estimates by direct counting. Here, let $\widehat{P}$ denote the computed estimate of the matrix $P$ and $\hat{M}$ be the estimate of the diagonal matrix $M$. We obtain a regret upper bound via an optimal frequency vector $\widehat{f}^*$ (see Step 5 in CONVEXPLORE).

Recall that for any vector $x$ (with non-negative components), the Hadamard exponentiation $\circ - 0.5$ leads to the vector $y = x^{\circ - 0.5}$ wherein $y_i = 1/\sqrt{x_i}$ for each component $i$. We next define a key parameter $\lambda$ that specifies the regret bound in Theorem 1 (below). At a high-level, parameter $\lambda$ captures the "exploration efficacy" in the MDP, that takes into account the transition probabilities $P$ and the exploration requirements $M$ at the intermediate layer. Identification of this parameter is a relevant technical contribution of our work; see Section C.1 for a detailed derivation of $\lambda$.

$$\lambda := \min_{\text{fq. vector} f} \left\| P M^{0.5} \left( P^\top f \right)^{\circ - 0.5} \right\|_\infty^2 \tag{3}$$

Furthermore, we will write $f^*$ to denote the optimal frequency vector in equation (3). Hence, with vector $\nu := P M^{0.5} (P^\top f^*)^{\circ - 0.5}$, we have $\lambda = \max_a \nu_a^2$. Note that Step 5 in CONVEXPLORE addresses an analogous optimization problem, albeit with the estimates $\widehat{P}$ and $\hat{M}$. Also, we show in Lemma 11 (see Section E in the supplementary material) that this optimization problem is convex and, hence, Step 5 admits an efficient implementation.

To understand the behaviour of $\lambda$, we first note that whenever the $m_i$ values at the contexts $i \in [k]$ are low, the $\lambda$ value is low. Specifically, the $m_i$ values can go as low as 2 (when the $q_j^i$s are all $\frac{1}{2}$), removing the dependence of $\lambda$ on $n$. The upper-bound on $\lambda$ is $nk$. We see this by first upper-bounding each $m_i$ by $n$. Then, note that whenever $\max_{a \in \mathcal{A}} P\{i|a\} \geq 1/k$, then $\exists f$ such that $P^\top f = u$ where $u = \{\frac{1}{k}, \ldots, \frac{1}{k}\}$. Now we can compute that $||P \cdot u^{\circ - 0.5}||_\infty^2 = k$, and thereby $\lambda < nk$; See footnote[2].

The following theorem that upper bounds the regret of CONVEXPLORE is the main result of the current work. The result requires the algorithm's time budget to be at least $T_0 := \widetilde{O}\left(N \max(m_i)/p_+^3\right)$

**Theorem 1.** Given number of rounds $T \geq T_0$ and $\lambda$ as in equation (3), CONVEXPLORE achieves regret

$$\text{Regret}_T \in \mathcal{O}\left( \sqrt{\max\left\{ \frac{\lambda}{T}, \frac{m_0}{T p_+} \right\} \log\left(NT\right)} \right)$$

Observe that $m_0/T p_+$ is independent of the number of contexts and interventions. Therefore $\lambda$ dominates when number of interventions at an intermediate context is large.

## 4 ANALYSIS OF THE LOWER BOUND

Since CONVEXPLORE solves an optimization problem, it is a priori unclear that a better algorithm may not provide a regret guarantee better than Theorem 1. In this section, we show that for a large class of instances, it is indeed the case that the regret guarantee we provide is optimal. We provide a lower bound on regret for a family of instances. For any number of contexts $k$, we show that there exist transition matrices $P$ and reward distributions $(\mathbb{E}[R_i \mid a])$ such that regret achieved by CONVEXPLORE (Theorem 1) is tight, up to log factors.

---

[2]$\lambda$ is upperbounded by kn, but is typically significantly smaller (as m may be much smaller than n).

**Theorem 2.** For any $q_j^i$ corresponding to causal variables at contexts $i \in [k]$, there exists a transition matrix $P$, and probabilities $q_j^0$ corresponding to causal variables $\{X_j^0\}_{j \in [n]}$, and reward distributions, such that the simple regret achieved by *any* algorithm is

$$\text{Regret}_T \in \Omega \left( \sqrt{\frac{\lambda}{T}} \right)$$

We provide the details of the proof of Theorem 2 in Section F in the supplementary material.

## 5 EXPERIMENTS

We first describe UNIFEXPLORE (Uniform Exploration Algorithm), the baseline algorithm that we compare CONVEXPLORE with. This is followed by a complete description of our experimental setup. Finally, we present and discuss our main results.

**Uniform Exploration (UnifExplore):** This algorithm uniformly explores the interventions in the instance. It first performs all the atomic interventions $a \in \mathcal{A}_0$ at the start state 0 in a round robin manner. On transitioning to any context $i \in [k]$, it performs atomic interventions $b \in \mathcal{A}_i$ in a round robin manner. **Note:** UNIFEXPLORE achieves a regret upperbounded by $\tilde{\mathcal{O}}(\sqrt{nk/T})$, which is also the optimal lower bound for non-causal algorithms. Hence it serves as a good comparison as it achieves an optimal non-causal simple regret. Moreover, in causal intervention contexts, we are unaware of any existing algorithm that could be adapted to serve as a baseline for our method.

**Setup:** We consider $k = 25$ intermediate contexts and a causal graphs with $n = 25$ variables ($2n + 1 = 51$ interventions) at each context. The rewards are distributed Bernoulli$(0.5 + \varepsilon)$ for intervention $X_1^1 = 1$ and Bernoulli$(0.5)$ otherwise where $\varepsilon = 0.3$ in the experiments. We set $m_i = m \ \forall i \in [k]$. As in experiments in prior work, we set $q_j^i = 0$ for $j \le m_i$ and 0.5 otherwise. Let $k = n$ here. At state 0, on taking action $a = do()$, we transition uniformly to one of the intermediate contexts. On taking action $do(X_i^0 = 1)$, we transition with probability $2/k$ to context $i$ and probability $1/k - 1/(k(k-1))$ to any of the other $k - 1$ contexts.

We perform two experiments in this setting. In the first one, we run CONVEXPLORE and UNIFEXPLORE for time horizon $T \in \{1000, \ldots, 25000\}$. In the second experiment, we run CONVEXPLORE and UNIFEXPLORE for a fixed time horizon $T = 25000$ with $\lambda$ varying in the set $\{50, 75, \ldots, 625\}$. To vary $\lambda$, we vary $m_i$ for the intermediate contexts in the set $\{2, 3, \ldots, 25\}$. We average the regret over 10000 runs for each setting. We use CVXPY (Diamond & Boyd (2016)) to solve the convex program at Step 5 in CONVEXPLORE.

**Results:** In Figure 4a, we compare the expected simple regret of CONVEXPLORE vs. UNIF-EXPLORE. Our plots indicate that CONVEXPLORE outperforms UNIFEXPLORE and its regret falls rapidly as $T$ increases. In Figure 4b, we plot the expected simple regret against $\lambda$ for CONVEXPLORE and UNIFEXPLORE that was obtained in Experiment 2, and empirically validate their relationship that was proved in Theorem 1.

Figure Figure 4b shows that CONVEXPLORE outperforms UNIFEXPLORE for a wide range of $\lambda$s. The only reason UNIFEXPLORE outperforms CONVEXPLORE at higher $\lambda$ values is due to higher constants for CONVEXPLORE. Specifically, reusing the data from first two phases of CONVEXPLORE in the subsequent phase may help CONVEXPLORE outperform UNIFEXPLORE even for higher $\lambda$ values. This highlights the applicability of CONVEXPLORE, specifically in instances wherein $\lambda$ is dominated by the other instance parameters. This also substantiates the relevance of using causal information; note that, by construction, CONVEXPLORE uses such information, whereas UNIFEXPLORE does not. For this instance, the worst case regret for UNIFEXPLORE is $\tilde{\mathcal{O}}(\sqrt{nk/T})$. In comparison, the worst case regret for CONVEXPLORE which is $\tilde{\mathcal{O}}(\sqrt{\lambda/T})$. While $\lambda \le nk$, the constant factors of UNIFEXPLORE may be smaller, and hence, for higher values of $\lambda$, we find that UNIFEXPLORE is competitive with CONVEXPLORE – as observed in Figure 4b.

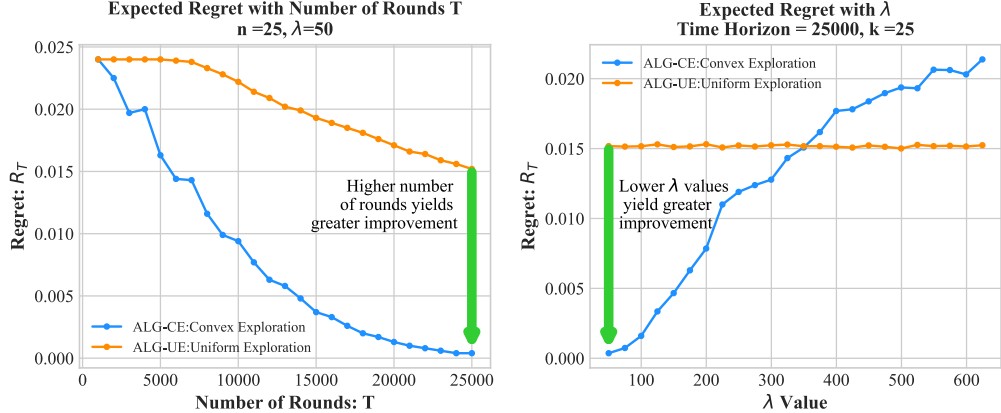

Figure 4: We plot the Simple Regret under CONVEXPLORE and UNIFEXPLORE. The figure on the left (4a) plots expected simple regret vs time, for the setup $n = 25$, $k = 25$, $\lambda = 50$, $\varepsilon = 0.3$ and $m = 2$ for all contexts. The figure on the right (4b) plots expected simple regret with $\lambda$. It was performed with the parameters: $T = 25000$, $k = 25$, $m_0 = 2$ and $\varepsilon = 0.3$.

## 6    CONCLUSION AND FUTURE WORK

We studied extensions of the causal contextual bandits framework to include adaptive context choice. This is an important problem in practice and the solutions therein have immediate practical applications. The setting of stochastic transition to a context accounted for non-trivial extensions from Subramanian & Ravindran (2022) who studied targeted interventions. We developed a Convex Exploration algorithm for minimizing simple regret under this setting. Furthermore, while Maiti et al. (2022) studied the simple causal bandit setting with unobserved confounders, our work addresses causal contextual bandits with adaptive contexts, under the same constraint of allowing unobserved confounders (assuming identifiability). We identified an instance dependent parameter $\lambda$, and proved that the regret of this algorithm is $\tilde{\mathcal{O}}(\sqrt{\frac{1}{T} \max\{\lambda, \frac{m_0}{p_+}\}})$. The current work also established that, for certain families of instances, this upper bound is essentially tight. Finally, we showed through experiments that our algorithm performs better than uniform exploration in a range of settings. We believe our method of converting the exploration in the causal contextual bandit setting is novel, and may have implications outside the causal setting as well.

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
