# SUPPLEMENTARY MATERIAL FOR ICLR 2024

## A    RELATED WORK

In our work, we draw from prior literature from causality as well as from multi-armed bandits. We will briefly cover these two in the following section.

### A.1    MULTI-ARMED BANDITS:

The stochastic Multi-Armed Bandit (MAB) setup is a standard model for studying the exploration-exploitation trade-off in sequential decision making problems (Kuleshov & Precup, 2014; Bubeck et al., 2012). Such trade-offs arise in several modern applications, such as ad placement, website optimization, recommendation systems, and packet routing (Bouneffouf et al., 2020) and are thus a central part of the theory relating to online learning (Slivkins et al., 2019; Lattimore & Szepesvári, 2020).

Traditional performance measures for MAB algorithms have focused on cumulative regret (Auer et al., 2002; Agrawal & Goyal, 2012; Auer & Ortner, 2010), as well as best-arm identification under the fixed confidence (Even-Dar et al., 2006) and fixed budget (Audibert et al., 2010) settings. In some settings however, one may be interested in optimizing the exploration phase. Another variant of regret that has been considered is the mini-max regret (Azar et al., 2017) which focuses on the worst case over all possible environments. However, as a metric for pure exploration in MABs, simple regret has been proposed as a natural performance criterion (Bubeck et al., 2009). In this setting, we allow for some period of exploration, after which the learner has to choose an arm. The simple regret is then evaluated as the difference between the average reward of the best arm and the average reward of the learner's recommendation. We focus on simple regret in this work.

Each of these performance metrics come with their own lower bounds (Orabona et al., 2012; Osband & Van Roy, 2016; Bubeck et al., 2012), which are naturally the benchmarks for any algorithms proposed. The lower bound on simple regret is known to be $\mathcal{O}(\sqrt{n/T})$ for a stochastic multi-armed bandit problem with $n$ arms. This bound is obtained from the lower bound for pure exploration provided by Mannor & Tsitsiklis (2004).

Note that, a naive approach to the causal bandit problem which simply treats an intervention on each of exponentially many combinations of the nodes as an arm, may thus incur an exponential regret. We now review some of the literature from Causality, which helps in addressing the causal aspects of the problem.

### A.2    CAUSALITY:

There are three broad threads in causality related to our work. These are causal graph learning, causal testing and causal bandits. We address relevant works in these areas below.

**Learning Causal Graphs:** Tian & Pearl (2002) laid the grounds for analysing functional functional constraints among the distributions of observed variables in a causal Bayesian networks. Similarly, Kang & Tian (2006) derive such functional constraints over interventional distributions. These two seminal works lead to a great interest in the problem of learning causal graphs.

There have been several studies that provide algorithms to recover the causal graphs from the conditional independence relations in observational data (Pearl & Verma, 1995; Spirtes et al., 2000; Ali et al., 2005; Zhang, 2008). Subsequent work considered the setting when both observational and interventional data are available (Eberhardt et al., 2005; Hauser & Bühlmann, 2014). Kocaoglu et al. (2017a) extend the causal graph learning problem to a budgeted setting. Shanmugam et al. (2015) uses interventions on sets of small size to learn the causal structure. Kocaoglu et al. (2017b) provide an efficient randomized algorithm to learn a causal graph with confounding variables.

**Testing over Bayesian networks:** Given sample access to an unknown Bayesian Network (Canonne et al., 2017), or Ising model (Daskalakis et al., 2019), one may wish to decide whether an unknown model is equal to a known fixed model, and analyse the sample complexity of this hypothesis test. Acharya et al. (2018) address this question by introducing the concept of covering interventions. These covering interventions allow us to understand the behaviour of multiple interventions (that are covered) simultaneously. We utilize the concept of covering interventions from Acharya et al. (2018) towards our question of finding the optimal intervention in a causal bandit. The area of reinforcement learning over causal bandits has also been studied in Zhang (2020).

Apart from these areas in causality, our primary problem of causal bandits have been addressed by Lattimore et al. (2016); Maiti et al. (2022); Sen et al. (2017a); Lu et al. (2020); Nair et al. (2021); Sen et al. (2017b); Lu et al. (2021; 2022); Varici et al. (2022); Xiong & Chen (2023). We detail these in the main Related Works Section 1.2.

## B    ALGORITHMS IN DETAIL

In this section, we outline the three algorithms that are used as helpers in CONVEXPLORE. The first that we outline now, Algorithm 2, would be used to estimate the transition probabilities out of context 0 on taking various actions.

---

**Algorithm 2** Estimate Transition Probabilities

---

1: **Input:** Time budget $T'$
2: **For** time $t \leftarrow \{1, \ldots, \frac{T'}{2}\}$ **do**
3:      Perform $do()$ at context 0. Transition to $i \in [k]$
4:      Count number of times context $i \in [k]$ is observed
5:      Update $\hat{q}_j^0 = \mathbb{P}\left\{X_j^0 = 1\right\}$
     **end**
6: Using $\hat{q}_j^0$s, estimate $m_0$ and the set $\mathcal{A}_{m_o}$. Estimate $\widehat{P}_{(a,i)} = \mathbb{P}[i \mid a] \quad \forall a \in \mathcal{A}_{m_0}^c$ and $i \in [k]$
7: **For** intervention $a \in \mathcal{A}_{m_o}$ at context 0
8:      **For** time $t \leftarrow \{1, \ldots \frac{T'}{2|\mathcal{A}_{m_0}|}\}$
9:          Perform $a \in \mathcal{A}_{m_o}$ and transition to some $i \in [k]$
10:          Count number of times context $i$ is observed
         **end**
     **end**
11: Estimate $\widehat{P}_{(a,i)} = \mathbb{P}[i \mid a]$ for each $a \in \mathcal{A}_{m_0}$ and contexts $i \in [k]$
12: **return** Estimated matrix $\widehat{P} = \left[\widehat{P}_{(a,i)}\right]_{i \in [k], a \in \mathcal{A}_0}$

---

[a] In the first half of time $T'/2$, we perform $do()$ at State 0.

[b] If $\mathcal{A}_0 := do() \cup \{X_j^0 = 0, X_j^0 = 1\}_{j \in [n]}$, we can find $m_0 \leq |\mathcal{A}_0|/2$ such that $\mathcal{A}_0 = \mathcal{A}_{m_0} \cup \mathcal{A}_{m_0}^c$ where the interventions in $\mathcal{A}_{m_0}^c$ are observed with probability more than $1/m_0$ and $|\mathcal{A}_{m_0}| = m_0$.

[c] For the interventions $a \in \mathcal{A}_{m_0}^c$, we can estimate $\widehat{P}_{(a,i)} = \mathbb{P}[i \mid a] \quad \forall i \in [k]$ in the first half.

[d] In the second half, we may intervene on the atomic interventions in $\mathcal{A}_{m_0}$ for time $T/(2m_0)$ each.

[e] Using observations of $a \in \mathcal{A}_{m_0}$, we estimate $\widehat{P}_{(a,i)} = \mathbb{P}[i \mid a] \quad \forall a \in \mathcal{A}_{m_0}$ and $i \in [k]$.

---

Next we outline two algorithms that estimate parameters for the other contexts $i \in [k]$. To estimate the causal parameters at all contexts $i \in [k]$, we use Algorithm 3. Then we will use Algorithm 4 to estimate the rewards on various interventions at the intermediate contexts.

For estimating the causal parameters, we use a variant of SRM-ALG, which estimates the causal observational threshold $m_i$, under the setting of unobserved confounders and identifiability. We note that even in the presence of general causal graphs with hidden variables, SRM-ALG is able to efficiently estimate the rewards of all the arms simultaneously using the observational arm pulls. As mentioned in Section 3 of Maiti et al. (2022), the challenge is to identify the optimal number of arms with bad estimates during the initial phase of the algorithm, such that these arms can be intervened upon at the later phase. The $q_i(x)$

parameter is the minimum conditional probability of $X = x$, given different configurations of the parents of $X$. Once we have these estimates, the remaining algorithm can proceed as per usual.

---

**Algorithm 3** Estimate Causal Parameters

---

1: **Input:** Frequency vector $\tilde{f}$ and time budget $T'$
2: Update $f(a) \leftarrow \frac{1}{2} \left( \tilde{f}(a) + \frac{1}{|\mathcal{A}_0|} \right) \quad \forall a \in \mathcal{A}_0$
3: **For** intervention $a \in \mathcal{A}_0$
4:      **For** time $t \leftarrow \{1, \ldots T' \cdot f(a)\}$
5:          Perform $a \in \mathcal{A}_0$ and transition to some $i \in [k]$.
6:          At context $i$, perform $do()$ and observe $X_j^i$s
7:          Update $\widehat{q}_j^i = \min_{\text{Parents}(X_j^i)} \mathbb{P} \left\{ X_j^i = 1 \mid \text{Parents}(X_j^i) \right\}$
     **end**
   **end**
8: Using $\widehat{q}_j^i$s, estimate $\widehat{m}_i$ values for each context $i \in [k]$
9: **return** $\hat{M}$, the diagonal matrix of the $\widehat{m}_i$ values

---

    [a]We choose actions $a \in \mathcal{A}_0$ such that we visit the contexts $i \in [k]$ approximately equally, in expectation.
    [b]On each visit to a context $i \in [k]$, we perform $do()$. From these we can estimate $q_i^j$ values, which may be used to estimate $m_i$ values.
    [c]Based on these $do()$ interventions at each context $i \in [k]$, we get estimates of $m_i$ and the intervention sets $\mathcal{A}_{m_i}$ such that (I) $|\mathcal{A}_{m_i}| = m_i$ and (II) interventions in $\mathcal{A}_{m_i}$ are observed with probability less than $1/m_i$.

Note that in Algorithm 4 there are two phases. In the first phase, we carry out estimates for interventions that have high probability of being observed on the $do()$ intervention. In the second phase, we specifically perform interventions which have not been observed often enough. This is similar to Algorithm 2 where we carry out the two phases of interventions at context 0.

**Algorithm 4** Estimate Rewards

1: **Input:** Optimal frequency $f^*$, min-max frequency $\tilde{f}$, and time budget $T'$
2: Set $f(a) \leftarrow \frac{1}{3}\left(f^*(a) + \tilde{f}(a) + \frac{1}{|\mathcal{A}_0|}\right) \ \ \forall a \in \mathcal{A}_0$
3: **For** intervention $a \in \mathcal{A}_0$ at context 0
4:     **For** time $t \leftarrow \{1, \ldots f(a) \cdot T'/2\}$
5:         Perform $a \in \mathcal{A}_0$. Transition to some $i \in [k]$. Perform $do()$ at context $i \in [k]$.
6:         Observe variables $X_j^i$'s and rewards $R_i$.
    **end**
  **end**
7: Find the set $\mathcal{A}_{m_i} \ \ \forall i \in [k]$ using $q_j^i$ estimates.
8: Estimate mean reward $\widehat{\mathcal{R}}_{(b,i)} = \mathbb{E}\left[R_i \mid b\right]$ for each $b \in \mathcal{A}_{m_i}^c$
9: **For** intervention $a \in \mathcal{A}_0$ at context 0
10:    **For** time $t \leftarrow \{1, \ldots f(a) \cdot T'/2\}$
11:       Perform $a \in \mathcal{A}_0$ and transition to some $i \in [k]$.
12:       Iteratively perform $b \in \mathcal{A}_{m_i}$. Observe $R_i$
    **end**
  **end**
13: Estimate mean reward $\widehat{\mathcal{R}}_{(b,i)} = \mathbb{E}\left[R_i \mid b\right]$ for each $b \in \mathcal{A}_{m_i}$
14: **return** $\widehat{\mathcal{R}} = \left[\widehat{\mathcal{R}}_{(b,i)}\right]_{i \in [k], b \in \mathcal{A}_i}$

---

[a]We perform the interventions in the ratio of $f^*$ which is the optimum given the $m_i$ values at the various contexts.

[b]In the first half we estimate rewards for the interventions $\mathcal{A}_{m_i}^c$ in the first half, and the interventions in $\mathcal{A}_{m_i}$ in the second half.

[c]Note that we round robin over the interventions $b \in \mathcal{A}_{m_i}$ across visits in the second half of the algorithm.

## C    Proof of Theorem 1

In this section, we restate Theorem 1 and provide its proof, along with all the lemmas that are used in the proof.

**Theorem.** Given number of rounds $T \geq T_0$ and $\lambda$ as in equation (3), ConvExplore achieves regret

$$\text{Regret}_T \in \mathcal{O}\left(\sqrt{\max\left\{\frac{\lambda}{T}, \frac{m_0}{Tp_+}\right\} \log\left(NT\right)}\right)$$

### C.1    Proof of Theorem 1

To prove the theorem, we analyze the algorithm's execution as falling under either *good event* or *bad event*, and tackle the regret under each.

**Definition 1.** We define five events, $E_1$ to $E_5$ (see Table 3), the intersection of which we call as *good event*, $E$, i.e., *good event* $E := \bigcap_{i \in [5]} E_i$. Furthermore, we define the *bad event* $F := E^c$.

Table 3: Table enumerating Good Events

| Event | Condition | Explanation |
|-------|-----------|-------------|
| $E_1$ | $\sum_{i=1}^{k} |\widehat{P}_{(a,i)} - P_{(a,i)}| \leq \frac{p_+}{3} \forall a \in \mathcal{A}_0$ | for every intervention $a \in \mathcal{A}_0$, the empirical estimate of transition probability in each of Algorithms 2, 3 and 4 is good, up to an absolute factor of $p_+/3$ |
| $E_2$ | $\widehat{m}_0 \in [\frac{2}{3}m_0, 2m_0]$ | our estimate for causal parameter $m_0$ for state 0 is relatively good in Algorithm 2. |
| $E_3$ | $\widehat{m}_i \in [\frac{2}{3}m_i, 2m_i] \quad \forall i \in [k]$ | our estimate for causal parameter $m_i$ for each context $i \in [k]$ is relatively good in Algorithm 3. |
| $E_4$ | $\sum_{i \in [k]} |\widehat{P}_{(a,i)} - P_{(a,i)}| \leq \zeta$, $\forall a \in \mathcal{A}_0$ | The error in estimated transition probability in Algorithm 2 sums to less than $\zeta$ where $\zeta := \sqrt{\frac{150m_0}{Tp_+} \log\left(\frac{3T}{k}\right)}$ |
| $E_5$ | $\left|\mathbb{E}\left[R_i \mid a\right] - \widehat{\mathcal{R}}_{(a,i)}\right| \leq \widehat{\eta}_i \;\; \forall i \in [k], a \in \mathcal{A}_i$ | The error in reward estimates in Algorithm 4 is bounded[3] by $\widehat{\eta}_i$ where $\widehat{\eta}_i = \sqrt{\frac{27\widehat{m}_i}{T(\widehat{P}^\top \widehat{f}^*)_i} \log\left(2TN\right)}$ |

Considering the estimates $\widehat{P}$ and $\hat{M}$, along with frequency vector[2] $\widehat{f}^*$ (computed in Step 5), we define random variable

$$\widehat{\lambda} := \left\|\widehat{P}\hat{M}^{1/2}\left(\widehat{P}^\top \widehat{f}^*\right)^{\circ - \frac{1}{2}}\right\|_\infty^2.$$

Note that $\widehat{\lambda}$ is a surrogate for $\lambda$. We will show that under the good event, $\widehat{\lambda}$ is close to $\lambda$ (Lemma 3).

Recall that $\text{Regret}_T := \mathbb{E}[\varepsilon(\pi)]$ and here the expectation is with respect to the policy $\pi$ computed by the algorithm. We can further consider the expected sub-optimality of the algorithm and the quality of the estimates (in particular, $\widehat{P}$, $\hat{M}$ and $\widehat{\lambda}$) under *good event* (E).

Based on the estimates returned at Step 5 of ConvExplore, either the *good event* holds, or we have the *bad event*. We obtain the regret guarantee by first bounding sub-optimality of policies computed under the *good event*, and then bound the probability of the *bad event*.

---

[3]Recall that $\widehat{f}^*$ denotes the optimal frequency vector computed in Step 5 of ConvExplore. Also, $(\widehat{P}^\top \widehat{f}^*)_i$ denotes the $i$th component of the vector $P^\top f^*$.

**Lemma 1.** For the optimal policy $\pi^*$, under the *good event* $(E)$, we have
$\sum_{i \in [k]} P_{(\pi^*(0),i)} \mathbb{E}\left[R_i \mid \pi^*(i)\right] - \sum \widehat{P}_{(\pi^*(0),i)} \widehat{\mathcal{R}}_{(\pi^*(i),i)} \leq \mathcal{O}\left(\sqrt{\max\{\widehat{\lambda}, m_0/p_+\}/T \log\left(NT\right)}\right)$

*Proof.* We add and subtract $\sum_{i \in [k]} P_{(\pi^*(0),i)} \widehat{\mathcal{R}}_{(\pi^*(i),i)}$ and reduce the expression on the left to: $\sum_{i \in [k]} P_{(\pi^*(0),i)}(\mathbb{E}\left[R_i \mid \pi^*(i)\right] - \widehat{\mathcal{R}}_{(\pi^*(i),i)}) + \sum_{i \in [k]} \widehat{\mathcal{R}}_{(\pi^*(i),i)}(P_{(\pi^*(0),i)} - \widehat{P}_{(\pi^*(0),i)})$.

We have: (a) $\widehat{\mathcal{R}}_{(\pi^*(i),i)} \leq 1$ (as rewards are bounded) (b) $\sum_{i \in [k]}|\widehat{P}_{(\pi^*(0),i)} - P_{(\pi^*(0),i)}| \leq \zeta$ (by $E_4$) and (c) $\left|\mathbb{E}\left[R_i \mid \pi^*(i)\right] - \widehat{\mathcal{R}}_{(\pi^*(i),i)}\right| \leq \widehat{\eta}_i$ (by $E_5$). The above expression is thus bounded above by $\sum_{i \in [k]} P_{(\pi^*(0),i)}\widehat{\eta}_i + \zeta$ Furthermore, it follows from $E_1$ (See Corollary 2 in Section D.1 in the supplementary material) that (component-wise) $P \leq \frac{3}{2}\widehat{P}$. Hence, the above-mentioned expression is bounded above by $\frac{3}{2} \sum_{i \in [k]} \widehat{P}_{(\pi^*(0),i)}\widehat{\eta}_i + \zeta$. Note that the definition of $\widehat{\lambda}$ ensures $\sum_{i \in [k]} \widehat{P}_{(\pi^*(0),i)}\widehat{\eta}_i = \mathcal{O}(\sqrt{\widehat{\lambda}/T \log(NT)})$. Further, $\zeta = \mathcal{O}(\sqrt{m_0/(Tp_+) \log(T/k)})$. Hence, $\sum_{i \in [k]} P_{(\pi^*(0),i)}\eta_i + \zeta = \mathcal{O}(\sqrt{\max\{\widehat{\lambda}, m_0/p_+\}/T \log\left(NT\right)})$, which establishes the lemma. $\square$

We now state another similar lemma for any policy $\widehat{\pi}$ computed under *good event*.

**Lemma 2.** Let $\widehat{\pi}$ be a policy computed by CONVEXPLORE under the *good event* $(E)$. Then, $\sum_{i \in [k]} \widehat{P}_{(\widehat{\pi}(0),i)} \widehat{\mathcal{R}}_{(\widehat{\pi}(i),i)} - \sum_{i \in [k]} P_{(\widehat{\pi}(0),i)}\mathbb{E}\left[R_i \mid \widehat{\pi}(i)\right] \leq \mathcal{O}\left(\sqrt{\max\{\widehat{\lambda}, m_0/p_+\}/T \log\left(NT\right)}\right)$

*Proof.* We can add and subtract $\sum_{i \in [k]} P_{(\widehat{\pi}(0),i)} \widehat{\mathcal{R}}_{(\widehat{\pi}(i),i)}$ to the expression on the left to get: $\sum_{i \in [k]} \widehat{\mathcal{R}}_{(\widehat{\pi}(i),i)}(\widehat{P}_{(\widehat{\pi}(0),i)} - P_{(\widehat{\pi}(0),i)}) + \sum_{i \in [k]} P_{(\widehat{\pi}(0),i)}(\widehat{\mathcal{R}}_{(\widehat{\pi}(i),i)} - \mathbb{E}\left[R_i \mid \widehat{\pi}(i)\right])$. Analogous to Lemma 1, one can show that this expression is bounded above by $\zeta + \sum_{i \in [k]} \frac{3}{2}\widehat{P}_{(\widehat{\pi}(0),i)}\widehat{\eta}_i = \mathcal{O}(\sqrt{\max\{\widehat{\lambda}, m_0/p_+\}/T \log\left(NT\right)})$. $\square$

We can also bound $\widehat{\lambda}$ to within a constant factor of $\lambda$.

**Lemma 3.** Under the *good event* $E$, we have $\widehat{\lambda} \leq 8\lambda$.

*Proof.* Event $E_1$ ensures that $\frac{2}{3}P \leq \widehat{P} \leq \frac{4}{3}P$ (see Corollary 2 in Appendix section D.1). In addition, note that event $E_3$ gives us $\widehat{M} \leq 2M$. From these observations we obtain the desired bound: $\widehat{\lambda} = \widehat{P}\widehat{M}^{0.5}(\widehat{P}^\top \widehat{f}^*)^{\circ-0.5} \leq \widehat{P}\widehat{M}^{0.5}(\widehat{P}^\top f^*)^{\circ-0.5} \leq 8PM^{0.5}(P^\top f^*)^{\circ-0.5} = 8\lambda$; here, the first inequality follows from the fact that $\widehat{f}^*$ is the minimizer of the $\widehat{\lambda}$ expression, and for the second inequality, we substitute the appropriate bounds of $\widehat{P}$ and $\widehat{M}$. $\square$

Recall that:

$$\pi^*(i) = \operatorname*{arg\,max}_{a \in \mathcal{A}_i} \mathbb{E}\left[R_i \mid a\right] \tag{4}$$

$$\pi^*(0) = \operatorname*{arg\,max}_{b \in \mathcal{A}_0}(\sum_{i=1}^k \mathbb{E}\left[R_i \mid \pi^*(i)\right] \cdot \mathbb{P}\{i \mid b\}) \tag{5}$$

We will now define $\varepsilon(\pi)$, denoting the sub-optimality of a policy $\pi$, as the difference between the expected rewards of $\pi^*$ and $\pi$. i.e. $\varepsilon(\pi) = \sum_{i=1}^k \mathbb{E}\left[R_i \mid \pi^*(i)\right] \cdot \mathbb{P}\{i \mid \pi^*(0)\} - \sum_{i=1}^k \mathbb{E}\left[R_i \mid \pi(i)\right] \cdot \mathbb{P}\{i \mid \pi(0)\}$.

**Corollary 1.** For any $\widehat{\pi}$ computed by CONVEXPLORE under *good event* $E$, $\varepsilon(\widehat{\pi}) = \mathcal{O}\left(\sqrt{\max\{\lambda, m_0/p_+\}/T \log\left(NT\right)}\right)$

*Proof.* Since CONVEXPLORE selects the optimal policy (maximizing rewards with respect to the estimates), $\sum \widehat{P}_{(\pi^*(0),i)}\widehat{\mathcal{R}}_{(\pi^*(i),i)} \leq \sum \widehat{P}_{(\widehat{\pi}(0),i)}\widehat{\mathcal{R}}_{(\widehat{\pi}(i),i)}$. Combining this

with Lemmas 1 and 2, we get $\sum_{i\in[k]} P_{(\pi^*(0),i)}\mathbb{E}[R_i\mid\pi^*(i)] - \sum_{i\in[k]} P_{(\widehat{\pi}(0),i)}\mathbb{E}[R_i\mid\widehat{\pi}(i)] = \mathcal{O}(\sqrt{\max\{\widehat{\lambda},m_0/p_+\}/T\log(NT)})$ under *good event*. The left-hand-side of this expression is equal to $\varepsilon(\widehat{\pi})$. Using Lemma 3, we get that $\varepsilon(\widehat{\pi}) = \mathcal{O}\left(\sqrt{\max\{\lambda,m_0/p_+\}/T\log(NT)}\right)$. □

Corollary 1 shows that under the *good event*, the (true) expected reward of $\pi^*$ and $\widehat{\pi}$ are within $\mathcal{O}\left(\sqrt{\max\{\lambda,m_0/p_+\}/T\log(NT)}\right)$ of each other. In Lemma 10 (see Section D.5 in the supplementary material) we will show [4] that $\mathbb{P}\{\bigcup_{i\in[5]}\neg E_i\} = \mathbb{P}\{F\} \leq 5k/T$ whenever $T \geq T_0$ [5].

The above-mentioned bounds together establish Theorem 1 (i.e., bound the regret of CONVEXPLORE): $\text{Regret}_T = \mathbb{E}[\varepsilon(\pi)] = \mathbb{E}[\varepsilon(\widehat{\pi})\mid E]\mathbb{P}\{E\} + \mathbb{E}[\varepsilon(\pi')\mid F]\mathbb{P}\{F\}$. Since the rewards are bounded between 0 and 1, we have $\varepsilon(\pi') \leq 1$, for all policies $\pi'$. But $\mathbb{P}\{E\} \leq 1$ giving us $\text{Regret}_T \leq \mathbb{E}[\varepsilon(\pi)\mid E] + \mathbb{P}\{F\}$. Therefore, Corollary 1 along with Lemma 10, leads to guarantee $\text{Regret}_T = \mathcal{O}\left(\sqrt{\max\{\lambda,m_0/p_+\}/T\log(NT)}\right) + 5k/T = \mathcal{O}\left(\sqrt{\max\{\lambda,m_0/p_+\}/T\log(NT)}\right)$

## D    BOUNDING THE PROBABILITY OF THE BAD EVENT

Recall that the *good event* corresponds to $\bigcap_{i\in5} E_i$ (see Definition 1). Write $F := \neg\left(\bigcap_{i\in5} E_i\right)$ and note that, for the regret analysis, we require an upper bound on $\mathbb{P}\{F\} = \mathbb{P}\{\neg(\bigcap_{i\in5} E_i)\} = \mathbb{P}\{\bigcup_{i\in5}\neg E_i\}$. Towards this, in this section we address $\mathbb{P}\{\neg E_i\}$, for each of the events $E_1$-$E_5$, and then apply the union bound.

### D.1    BOUND ON $\neg E_1$

The next lemma upper bounds the probability of $\neg E_1$.

**Lemma 4.** In each of Algorithms 2, 3 and 4 and for all interventions $a \in \mathcal{A}_0$, we have $\mathbb{P}\{\neg E_1\} = \mathbb{P}\left\{\sum_{i=1}^k|\widehat{P}_{(a,i)} - P_{(a,i)}| > \frac{p_+}{3}\right\} < \frac{k}{T}$ whenever $T \geq \max\left\{\frac{1620N}{p_+^3}, \frac{2025N}{p_+^2}\log\left(\frac{9NT}{k}\right)\right\}$.

*Proof.* On performing any intervention $a \in \mathcal{A}_0$ at context 0, the intermediate context that we visit follows a multinomial distribution. Hence, we can apply Devroye's inequality (for multinomial distributions) to obtain a concentration guarantee; we state the inequality next in our notation.

**Lemma 5** (Restatement of Lemma 3 in Devroye (1983)). Let $T_a$ be the number of times intervention $a \in \mathcal{A}_0$ is performed in context 0. Then, for any $\eta > 0$ and any $T_a \geq \frac{20s}{\eta^2}$, we have $\mathbb{P}\left\{\sum_{i=1}^k|\widehat{P}_{(a,i)} - P_{(a,i)}| > \eta\right\} \leq 3\exp\left(-\frac{T_a\eta^2}{25}\right)$. Here, $s$ is the support of the distribution (i.e., the number of contexts that can be reached from $a$ with a nonzero probability).

Note that each intervention $a \in \mathcal{A}_0$ is performed at least $T_a = \frac{T}{9N}$ times across Algorithms 2, 3 and 4. Setting $\eta = \frac{p_+}{3}$ and $T_a = \frac{T}{9N}$ above, we get that for each intervention $a \in \mathcal{A}_0$, in each subroutine, $\mathbb{P}\left\{\sum_{i=1}^k|P_{(a,i)} - \widehat{P}_{(a,i)}| > \frac{p_+}{3}\right\} \leq 3\exp\left(-\frac{Tp_+^2}{9N\cdot9\cdot25}\right) = 3\exp\left(-\frac{Tp_+^2}{2025N}\right)$.

Note that to apply the inequality, we require $\frac{T}{9N} \geq \frac{180s}{p_+^2}$, i.e., $T \geq \frac{1620sN}{p_+^2}$. In the current context, the support size $s$ is at most $\frac{1}{p_+}$; this follows from the fact that on performing any intervention $a \in \mathcal{A}_0$, at most $\frac{1}{p_+}$ contexts can have $P_{(a,i)} \geq p_+$. Hence, the requirement reduces to $T \geq \frac{1620N}{p_+^3}$.

---

[4] Recall that, by definition, $F = E^c$.

[5] $T_0$ as defined in Lemma 10 in Section D.5 in the supplementary material.

Next, we union bound the probability over the $N$ interventions (at state $0$) and the three subroutines, to obtain that, for any intervention $a \in \mathcal{A}_0$ and in any subroutine, $\mathbb{P}\left\{\sum_{i=1}^{k} |P_{(a,i)} - \widehat{P}_{(a,i)}| > \frac{p_+}{3}\right\} \le 3N \cdot 3 \exp\left(-\frac{Tp_+^2}{2025N}\right) = 9N \exp\left(-\frac{Tp_+^2}{2025N}\right)$.

Note that $9N \exp\left(-\frac{Tp_+^2}{2025N}\right) \le \frac{k}{T}$, for any $T \ge \frac{2025N}{p_+^2} \log\left(\frac{9NT}{k}\right)$. Hence, for any $T \ge \max\left\{\frac{1620N}{p_+^3}, \frac{2025N}{p_+^2} \log\left(\frac{9NT}{k}\right)\right\}$, we have $\mathbb{P}[\neg E_1] \le 9N \exp\left(-\frac{Tp_+^2}{2025N}\right) \le \frac{k}{T}$. This completes the proof of the lemma. $\qquad\square$

We state below a corollary which provides a multiplicative bound on $\widehat{P}$ with respect to $P$, complementing the additive form of $E_1$.

**Corollary 2.** Under event $E_1$, we have $\frac{2}{3}P_{(a,i)} \le \widehat{P}_{(a,i)} \le \frac{4}{3}P_{(a,i)}$, for all interventions $a \in \mathcal{A}_0$ and contexts $i \in [k]$.

*Proof.* Event $E_1$ ensures that $\sum_{i=1}^{k} |\widehat{P}_{(a,i)} - P_{(a,i)}| \le \frac{p_+}{3}$, for each interventions $a \in \mathcal{A}_0$ and contexts $i \in [k]$. This, in particular, implies that for each intervention $a \in \mathcal{A}_0$ and context $i \in [k]$ the following inequality holds: $|\widehat{P}_{(a,i)} - P_{(a,i)}| \le \frac{p_+}{3}$. Note that if $P_{(a,i)} = 0$, then the algorithm will never observe context $i$ with intervention $a$, i.e., in such a case $\widehat{P}_{(a,i)} = P_{(a,i)} = 0$. For the nonzero $P_{(a,i)}$s, recall that (by definition), $p_+ = \min\{P_{(a,i)} \mid P_{(a,i)} > 0\}$. Therefore, for any nonzero $P_{(a,i)}$, the above-mentioned inequality gives us $|\widehat{P}_{(a,i)} - P_{(a,i)}| \le \frac{1}{3}P_{(a,i)}$. Equivalently, $\widehat{P}_{(a,i)} \le \frac{4}{3}P_{(a,i)}$ and $\widehat{P}_{(a,i)} \ge \frac{2}{3}P_{(a,i)}$. Therefore, for all $P_{(a,i)}$s the corollary holds. $\qquad\square$

### D.2    Bound on Events $\neg E_2$ and $\neg E_3$

In this section, we bound the probabilities that our estimated $\widehat{m}_i$s are far away from the true causal parameters $m_i$s.

**Lemma 6.** For any $T \ge 144m_0 \log\left(\frac{TN}{k}\right)$, in Algorithm 2, $\mathbb{P}[\neg E_2] = \mathbb{P}\left\{\widehat{m}_0 \notin [\frac{2}{3}m_0, 2m_0]\right\} \le \frac{k}{T}$.

*Proof.* We allocate time $\frac{T}{3}$ to Algorithm 2. Lemma 8 of Lattimore et al. (2016) ensures that, for any $\delta \in (0,1)$ and $\frac{T}{3} \ge 48m_0 \log(\frac{N}{\delta})$, we have $\widehat{m}_0 \in [\frac{2}{3}m_0, 2m_0]$, with probability at least $(1 - \delta)$. Setting $\delta = \frac{k}{T}$, we get the required probability bound. $\qquad\square$

Next, we address $\mathbb{P}\{\neg E_3 \mid E_1\}$.

**Lemma 7.** For any $T \ge \frac{648 \max(m_i)N}{p_+} \log(2NT)$, in each of Algorithms 3 and 4, we have $\mathbb{P}\left\{\exists i \in [k], \quad \widehat{m}_i \notin [\frac{2}{3}m_i, 2m_i] \mid E_1\right\} \le \frac{k}{T}$.

*Proof.* Fix any reachable context $i \in [k]$. Corresponding to such a context, there exists an intervention $\alpha \in \mathcal{A}_0$ such that $P_{(\alpha,i)} \ge p_+$. Event $E_1$ (Corollary 2) implies that $\widehat{P}_{(\alpha,i)} \ge \frac{2}{3}P_{(\alpha,i)} \ge \frac{2}{3}p_+$.

Now, write $T_i$ to denote the number of times context $i \in [k]$ is visited by the Algorithms 3 and 4. Recall that in the subroutines we estimate $\widehat{P}_{(\alpha,i)}$ by counting the number of times context $i$ was reached and simultaneously intervention $\alpha$ observed. Furthermore, note that we allocate to every intervention at least $\frac{T}{9N}$ time (See Steps 2 in both the subroutines). In particular, intervention $\alpha$ was necessarily observed $\frac{T}{9N}$ times. Therefore, $\widehat{P}_{(a,i)} \le \frac{T_i}{\left(\frac{T}{9N}\right)}$.

This inequality leads to a useful lower bound: $T_i \ge \frac{T}{9N} P_{(a,i)} \ge T\frac{2p_+}{27N}$.

We now restate Lemma 8 from Lattimore et al. (2016): Let $T_i$ be the number of times context $i \in [k]$ is observed. Then, $\mathbb{P}\left\{\widehat{m}_i \notin [\frac{2}{3}m_i, 2m_i]\right\} \leq 2N \exp\left(-\frac{T_i}{48m_i}\right)$.

Since $T_i \geq \frac{2Tp_+}{27N}$, this guarantee of Lattimore et al. (2016) corresponds to $\mathbb{P}\left\{\widehat{m}_i \notin [\frac{2}{3}m_i, 2m_i]\right\} \leq 2N \exp\left(-\frac{Tp_+}{648Nm_i}\right) \leq 2N \exp\left(-\frac{Tp_+}{648N\max(m_i)}\right)$.

Union bounding over all contexts $i \in [k]$ and the two Algorithms 3 and 4, we obtain $\mathbb{P}\left\{\exists i \in [k] \text{ in Algorithms 3, 4 with } \widehat{m}_i \notin [\frac{2}{3}m_i, 2m_i]\right\} \leq 2Nk \exp\left(-\frac{Tp_+}{648N\max(m_i)}\right)$. Finally, substituting the value of $T \geq \frac{648\max(m_i)N}{p_+}\log(2NT)$, gives us $\mathbb{P}\left\{\exists i \in [k] \text{ in Algorithms 3, 4 with } \widehat{m}_i \notin [\frac{2}{3}m_i, 2m_i]\right\} \leq 2Nk \exp\left(-\frac{p_+}{648N\max(m_i)} \cdot \left[\frac{648\max(m_i)N}{p_+}\log(2NT)\right]\right) = \frac{k}{T}$. This completes the proof. $\square$

### D.3 BOUND ON $E_4$:

The following lemma provides an upper bound for $\mathbb{P}\{\neg E_4 \mid E_2\}$.

**Lemma 8.** Let $\zeta := \sqrt{\frac{150m_0}{Tp_+}\log\left(\frac{3T}{k}\right)}$. Then, $\mathbb{P}\{\neg E_4 \mid E_2\} = \mathbb{P}\left\{\sum_{i \in [k]}\left|P_{(a,i)} - \widehat{P}_{(a,i)}\right| > \zeta \Big| E_2\right\} \leq \frac{k}{T}$.

*Proof.* As in the proof of Lemma 4, we will use Devroye's inequality. Write $T_a$ to denote the number of times intervention $a \in \mathcal{A}_0$ is observed (in state 0) in Algorithm 2. For any $\eta \in (0,1)$ and with $T_a \geq \frac{20s}{\eta^2}$, Devroye's inequality gives us $\mathbb{P}\left\{\sum_{i=1}^{k}|\widehat{P}_{(a,i)} - P_{(a,i)}| > \eta\right\} \leq 3\exp\left(-\frac{T_a\eta^2}{25}\right)$. Here, $s$ is the size of the support of the multinomial distribution.

We first show that $T_a$ is sufficiently large, for each intervention $a \in \mathcal{A}_0$. Recall that we allocate time $\frac{T}{3}$ to Algorithm 2. Furthermore, we observe each intervention in state 0, at least $\frac{T}{3\widehat{m}_0}$ times, either as part of the do-nothing intervention or explicitly in Step 9 of Algorithm 2. Now, event $E_2$ ensures that $\widehat{m}_0 \in [\frac{2}{3}m_0, 2m_0]$. Hence, each intervention $a \in \mathcal{A}_0$ is observed $T_a \geq \frac{T}{3\widehat{m}_0} \geq \frac{T}{3 \cdot 2m_0} = \frac{T}{6m_0}$ times.

Substituting this inequality for $T_a$ in the above-mentioned probability bound, we obtain $\mathbb{P}\left\{\sum_{i=1}^{k}|\widehat{P}_{(a,i)} - P_{(a,i)}| > \eta\right\} \leq 3\exp\left(-\frac{T\eta^2}{150m_0}\right)$ when $T \geq \frac{120sm_0}{\eta^2}$. As observed in Lemma 4, the support size $s$ is at most $\frac{1}{p_+}$. Therefore, the requirement on $T$ reduces to $T \geq \frac{120m_0}{\eta^2 p_+}$.

Setting $\eta = \sqrt{\frac{150m_0}{Tp_+}\log\left(\frac{3T}{k}\right)}$ gives us

$$\mathbb{P}\left\{\sum_{i=1}^{k}|\widehat{P}_{(a,i)} - P_{(a,i)}| > \sqrt{\frac{150m_0}{Tp_+}\log\left(\frac{3T}{k}\right)}\right\} \leq 3\exp\left(\frac{-T}{150m_0}\left[\sqrt{\frac{150m_0}{Tp_+}\log\left(\frac{3T}{k}\right)}\right]^2\right)$$

$$\leq \frac{k}{T}.$$

Therefore $\mathbb{P}\left\{\sum_{i=1}^{k}|\widehat{P}_{(a,i)} - P_{(a,i)}| > \eta\right\} \leq \frac{k}{T}$, and this probability bound requires $T \geq \frac{120m_0}{\eta^2 p_+}$. That is, $\eta \geq \sqrt{\frac{120m_0}{Tp_+}}$. This inequality is satisfied by our choice of $\eta$. Hence, the lemma stands proved. $\square$

### D.4 BOUND ON $\neg E_5$

The next lemma bounds $\mathbb{P}\{\neg E_5 \mid E_1, E_3\}$.

**Lemma 9.** Let $\widehat{\eta}_i = \sqrt{\frac{27\widehat{m}_i}{T(\widehat{P}^\top \widehat{f}^*)_i} \log (2TN)}$. Then, $\mathbb{P}\{\neg E_5 \mid E_3, E_1\} \leq \frac{k}{T}$. In other words:

$$\mathbb{P}\left\{\exists i \in [k] \text{ and } a \in \mathcal{A}_i \text{ such that } \left|\mathbb{E}\left[R_i \mid a\right] - \widehat{\mathcal{R}}_{(a,i)}\right| > \widehat{\eta}_i \mid E_3, E_1\right\} \leq \frac{k}{T}$$

.

*Proof.* For intermediate contexts $i \in [k]$, we denote the realization of the causal parameters $m_i$ and the transition probabilities $P$ in Algorithm 4, as $\widetilde{m}_i$ and $\widetilde{P}$, respectively. The estimates in the previous subroutines are denoted by $\widehat{m}_i$ and $\widehat{P}$.

Event $E_1$ gives us $P_{(a,i)} \in [\frac{3}{4}\widehat{P}_{(a,i)}, \frac{3}{2}\widehat{P}_{(a,i)}]$ and $\widetilde{P}_{(a,i)} \in [\frac{2}{3}P_{(a,i)}, \frac{4}{3}P_{(a,i)}]$. Hence, the estimates across the subroutines are close enough: $\widetilde{P}_{(a,i)} \in [\frac{1}{2}\widehat{P}_{(a,i)}, 2\widehat{P}_{(a,i)}]$. Similarly, event $E_3$ gives us $\widetilde{m}_i \in [\frac{1}{3}\widehat{m}_i, 3\widehat{m}_i]$.

Write $\widetilde{T}_i$ to denote the number of times context $i \in [k]$ was visited in Algorithm 4. For all contexts $i \in [k]$, we first establish a useful lower bound on $\widetilde{T}_i$, under events $E_1$ and $E_3$. The relevant observation here is that the estimate $\widetilde{P}_{(\alpha,i)}$ was computed in Algorithm 4 by counting the number of times context $i$ was visited with intervention $\alpha \in \mathcal{A}_0$ (at state 0). By construction, in Algorithm 4 each intervention $\alpha \in \mathcal{A}_0$ was performed at least $\frac{\widehat{f}^*_\alpha}{3}\frac{T}{3}$ times. Furthermore, given that $\widetilde{P}_{(\alpha,i)}$ was computed via the visitation count, we get that context $i$ is visited with intervention $\alpha \in \mathcal{A}_0$ at least $\widetilde{P}_{(\alpha,i)}\frac{T\widehat{f}^*_\alpha}{9}$ times. Therefore, $\widetilde{T}_i \geq \sum_{\alpha \in \mathcal{A}_0} \widetilde{P}_{(\alpha,i)}\frac{T\widehat{f}^*_\alpha}{9} = \frac{T}{9}(\widetilde{P}^\top \widehat{f}^*)_i \geq \frac{T}{18}(\widehat{P}^\top \widehat{f}^*)_i$. Here, the last inequality follows from the above-mentioned proximity between $\widehat{P}$ and $\widetilde{P}$.

Now, note that, at each context $i \in [k]$, Algorithm 4 (by construction) observes every intervention $a \in \mathcal{A}_i$ at least $\frac{\widetilde{T}_i}{\widetilde{m}_i}$ times. Write $\widetilde{T}_{(a,i)}$ to denote the number of times intervention $a \in \mathcal{A}_i$ is observed in this subroutine. Hence,

$$\widetilde{T}_{(a,i)} \geq \frac{\widetilde{T}_i}{\widetilde{m}_i} \geq \frac{1}{\widetilde{m}_i}\frac{T}{18}(\widehat{P}^\top \widehat{f}^*)_i \geq \frac{1}{3\widehat{m}_i}\frac{T}{18}(\widehat{P}^\top \widehat{f}^*)_i \tag{6}$$

For each context $i \in [k]$ and intervention $a \in \mathcal{A}_i$, define the event $\neg E_5(a,i)$ as $|\mathbb{E}\left[R_i \mid a\right] - \widehat{\mathcal{R}}_{(a,i)}| > \widehat{\eta}_i$. Hoeffding's inequality gives us $\mathbb{P}\{\neg E_5(a,i) \mid E_1, E_3\} \leq 2\exp\left(-2\widetilde{T}_{(a,i)}\widehat{\eta}_i^2\right) \leq 2\exp\left(-T\frac{(\widehat{P}^\top \widehat{f}^*)_i\widehat{\eta}_i^2}{27\widehat{m}_i}\right)$. The last inequality is obtained by substituting Equation 6.

Recall that $\widehat{\eta}_i = \sqrt{\frac{27\widehat{m}_i}{T(\widehat{P}^\top \widehat{f}^*)_i} \log (2TN)}$. Hence, the previous inequality corresponds to $\mathbb{P}\{\neg E_5(a,i) \mid E_1, E_3\} \leq 2\exp\left(-T\frac{(\widehat{P}^\top \widehat{f}^*)_i}{27\widehat{m}_i} \cdot \left[\sqrt{\frac{27\widehat{m}_i}{T(\widehat{P}^\top \widehat{f}^*)_i} \log (2TN)}\right]^2\right) = \frac{1}{TN}$.

Note that $\neg E_5 = \bigcup_{i \in [k]} \bigcup_{a \in \mathcal{A}_i} E_5(a,i)$. Taking a union bound over all contexts $i \in [k]$ and interventions $a \in \mathcal{A}_i$, we obtain $\mathbb{P}\{\neg E_5 \mid E_1, E_3\} \leq \frac{kN}{TN} = \frac{k}{T}$. This completes the proof. □

### D.5 BOUND ON *bad event* (F):

Write $T_0 := \mathcal{O}\left(\frac{N\max(m_i)}{p_+^3} \log (2NT)\right) = \widetilde{O}\left(\frac{N\max(m_i)}{p_+^3}\right)$.

**Lemma 10.** $\mathbb{P}\{F\} \leq \frac{5k}{T}$ for any $T > T_0$.

*Proof.* We summarize the statements of Lemmas 4, 6, 7, 8 and 9 as follows. When $T \geq T_0$ where $T_0 = \max\left\{\frac{1620N}{p_+^3}, \frac{2025N}{p_+^2} \log \left(\frac{9NT}{k}\right), 144m_0 \log \left(\frac{Tn}{k}\right), \frac{864\max(m_i)N}{p_+} \log (2nT)\right\} = \mathcal{O}\left(\frac{N\max(m_i)}{p_+^3} \log (2NT)\right)$, we obtain $\mathbb{P}\{F\} = \mathbb{P}\left\{\left[\bigcup_{i \in [5]} \neg E_i\right]\right\} \leq \mathbb{P}\{\neg E_1\} + \mathbb{P}\{\neg E_2\} + \mathbb{P}\{\neg E_3 \mid E_1\} + \mathbb{P}\{\neg E_4 \mid E_2\} + \mathbb{P}\{\neg E_5 \mid E_3, E_1\} \leq \frac{5k}{T}$. □

# E   Nature of the Optimization Problem

**Proposition E.1.** Let $\tilde{f} = \arg\max_{\text{fq. vector} f} \min_{\text{contexts } [k]} \widehat{P}^\top f$. Then, finding $\tilde{f}$ is an LP

*Proof.* We rewrite the above $\max_{\text{fq. vector} f} \min_{i \in [k]}(\cdot)$ as a simpler program:

$$\max_f \quad z$$

$$\text{subject to} \quad \widehat{P}_1^\top f \geq z$$

$$\cdots$$

$$\widehat{P}_N^\top f \geq z$$

$$f \cdot \mathbb{1} = 1$$

$$f \succeq 0$$

Where $N = |\mathcal{A}_0|$. This is equivalent to the standard form of a linear program, and hence is an LP. $\square$

**Lemma 11.** $\min_{\text{fq. vector} f} \max_{\text{interventions } \mathcal{A}_0} \widehat{P}\hat{M}^{\frac{1}{2}} \left[\widehat{P}^\top f\right]^{\circ - \frac{1}{2}}$ is a convex optimization problem

*Proof.* First we write the min-max in terms of a single minimization. First let us use the shorthand $A := \widehat{P}\hat{M}^{\frac{1}{2}}$ and $\{A_1, \ldots, A_N\}$ (where $N := |\mathcal{A}_0|$) denote the rows of the matrix

$$\mathbf{OPT} : \min_f \quad z$$

$$\text{subject to} \quad A_1 \cdot \left[\widehat{P}^\top f\right]^{\circ - \frac{1}{2}} \leq z$$

$$\cdots$$

$$A_N \cdot \left[\widehat{P}^\top f\right]^{\circ - \frac{1}{2}} \leq z \qquad (7)$$

$$f \cdot \mathbb{1} = 1$$

$$f \succeq 0$$

**Proposition E.2.** For any $a \in \mathbb{R}_+$, the function $g(x) := ax^{-\frac{1}{2}}$ is convex in $x$.

*Proof.* We observe that the second derivative is positive. $\square$

**Proposition E.3.** The constraint equations of OPT are convex in $f$

*Proof.* Consider the first constraint of the problem. We can simplify this to get $\sum_{i \in [k]} \frac{A_{1i}}{\sqrt{\widehat{P}(*,i)^\top f}}$.

Note that the $i$th term in the summand (i.e, $\frac{A_{1i}}{\sqrt{\widehat{P}(*,i)^\top f}}$) is of the form $f(x) = c(v^\top x)^{-\frac{1}{2}}$ for some $c \in \mathbb{R}_+$ and $v \in \mathbb{R}_+^N$. Let $x_1, x_2 \in \mathbb{R}^N$ be any two vectors, and scalar $\lambda \in [0, 1]$. We wish to show that $f(\lambda x_1 + (1 - \lambda)x_2) \leq \lambda f(x_1) + (1 - \lambda)f(x_2)$.

We have $f(\lambda x_1 + (1 - \lambda)x_2) = c(v^\top(\lambda x_1 + (1 - \lambda)x_2))^{-\frac{1}{2}} = c(\lambda v^\top x_1 + (1 - \lambda)v^\top x_2)^{-\frac{1}{2}}$

But $ax^{-\frac{1}{2}}$ is convex as per Proposition E.2. Therefore $c(\lambda v^\top x_1 + (1 - \lambda)v^\top x_2)^{-\frac{1}{2}} \leq \lambda c(v^\top x_1)^{-\frac{1}{2}} + (1 - \lambda)c(v^\top x_2)^{-\frac{1}{2}} = \lambda f(x_1) + (1 - \lambda)f(x_2)$, as required.

Since $\frac{A_{1i}}{\sqrt{\widehat{P}(*,i)^\top f}}$ is convex, the sum $\sum_{i \in [k]} \frac{A_{1i}}{\sqrt{\widehat{P}(*,i)^\top f}}$ is convex as well. Similarly, all the other constraints are also convex. $\square$

Since the constraints are convex in $f$ and the objective is linear, OPT is convex. $\qquad\square$

## F    Lower Bounds

This section establishes Theorem 2. We will identify a collection of instances for causal bandits with intermediate feedback and show that, for any given algorithm $\mathcal{A}$, there exists an instance in this collection for which $\mathcal{A}$'s regret is $\Omega\left(\sqrt{\frac{\lambda}{T}}\right)$.

First we describe the collection of instances and then provide the proof.

For any integer $k > 1$, consider $n = (k-1)$ causal variables at each context $i \in \{0, 1, \ldots, k\}$. The transition matrix $P$ is set to be deterministic. Specifically, for each $i \in [n]$, we have $\mathbb{P}\{i \mid do(X_i^0 = 1)\} = 1$. For all other interventions at context 0, we transition to context k with probability 1. Such a transition matrix can be achieved by setting $q_i^0 = 0$ for all $i \in [k-1]$. As before, the total number of interventions $N := 2n + 1 = 2k - 1$.

Now consider a family of $Nk + 1$ instances[6] $\{\mathcal{F}_0\} \cup \{\mathcal{F}_{(a,i)}\}_{i \in [k], a \in \mathcal{A}_i}$. Here, $\mathcal{F}_0$ and each $\mathcal{F}_{(a,i)}$ is an instance of a causal bandit with intermediate feedback with the above-mentioned transition probabilities. The instances differ in the rewards at the intermediate contexts. In particular, in instance $\mathcal{F}_0$, we set the reward distributions such that $\mathbb{E}[R_i \mid a] = \frac{1}{2}$ for all contexts $i \in [k]$ and interventions $a \in \mathcal{A}_i$. For each $i \in [k]$ and $a \in \mathcal{A}_i$, instance $\mathcal{F}_{(a,i)}$ differs from $\mathcal{F}_0$ only at context $i$ and for intervention $a$. Specifically, by construction, we will have $\mathbb{E}[R_i \mid a] = \frac{1}{2} + \beta$, for a parameter $\beta > 0$. The expected rewards under all other interventions will be $1/2$, the same as in $\mathcal{F}_0$.

Given any algorithm $\mathcal{A}$, we will consider the execution of $\mathcal{A}$ over all the instances in the family. The execution of algorithm $\mathcal{A}$ over each instance induces a trace, which may include the realized transition probabilities $\widehat{P}$, the realized variable probabilities $\widehat{q}_j^i$ for $i \in [k]$ and $j \in [n]$ and the corresponding $\widehat{m}_i$s, and the realized rewards $\widehat{\mathcal{R}}$. Each of such realizations (random variables) has a corresponding distribution (over many possible runs of the algorithm). We call the measures corresponding to these random variables under the instances $\mathcal{F}_0$ and $\mathcal{F}_{(a,i)}$ as $\mathcal{P}_0$ and $\mathcal{P}_{(a,i)}$, respectively.

### F.1    Proof of Theorem 2

For any algorithm $\mathcal{A}$ and given time budget $T$, we first consider the $\mathcal{A}$'s execution over instance $\mathcal{F}_0$. As mentioned previously, $\mathcal{P}_0$ denotes the trace distribution induced by the algorithm for $\mathcal{F}_0$. In particular, write $r_i$ to denote the expected number of times context $i$ is visited, $r_i := \mathbb{E}_{\mathcal{P}_0}[\text{state } i \text{ is visited}]/T$.

Recall that $m_i := \max\{j \mid q_{(j)}^i < \frac{1}{j}\}$ and $\mathcal{A}_{m_i} := \{do(X_{(j)}^i = 1) \mid q_{(j)}^i < \frac{1}{j}\}$, where the Bernoulli probabilities of the variables at context $i$ are sorted to satisfy $q_{(1)}^i \leq q_{(2)}^i \leq \cdots \leq q_{(n)}^i$. Note that these definitions do not depend on the algorithm at hand. The algorithm, however, may choose to perform different interventions different number of times. Write $N_{(a,i)}$ to denote the expected (under $\mathcal{P}_0$) number of times intervention $a$ is performed by the algorithm at context $i$. Furthermore, let random variable $T_{(a,i)}$ denote the number of times intervention $a$ is observed at context $i$. Hence, $\mathbb{E}_{\mathcal{P}_0}[T_{(a,i)}]$ is the expected number of times intervention $a$ is observed[7].

Using the expected values for algorithm $\mathcal{A}$ and instance $\mathcal{F}_0$, we define a subset of $\mathcal{A}_{m_i}$ as follows: $\mathcal{J}_i := \left\{ a \in \mathcal{A}_{m_i} : N_{(a,i)} \leq 2\frac{T r_i}{m_i} \right\}$. The following proposition shows that the size of $\mathcal{J}_i$ is sufficiently large.

---

[6]Note the change in notation. We used the term $\mathcal{F}_{i,j}$ instead of $\mathcal{F}_{(a,i)}$ in the main paper. This has been amended in a later version of the main paper.

[7]Note that $a$ can be observed while performing the do-nothing intervention. Also, the expected value $N_{(a,i)}$ accounts for the number of times $a$ is explicitly performed and not just observed.

**Proposition F.1.** The set $\mathcal{J}_i$ is non-empty. In particular,

$$m_i/2 \leq |\mathcal{J}_i| \leq m_i.$$

*Proof.* The upper bound on the size of subset $\mathcal{J}_i$ follows directly from its definition: since $\mathcal{J}_i \subseteq I_{m_i}$ we have $|\mathcal{J}_i| \leq |\mathcal{A}_{m_i}| = m_i$.

For the lower bound on the size of $\mathcal{J}_i$, note that $Tr_i$ is the expected number of times context $i$ is visited by the algorithm. Therefore,

$$\sum_{a \in \mathcal{A}_{m_i}} N_{(a,i)} \leq Tr_i \tag{8}$$

Furthermore, by definition, for each intervention $b \in \mathcal{A}_{m_i} \setminus \mathcal{J}_i$ we have $N_{(b,i)} \geq \frac{2Tr_i}{m_i}$. Hence, assuming $|\mathcal{A}_{m_i} \setminus \mathcal{J}_i| > \frac{m_i}{2}$ would contradict inequality (8). This observation implies that $|\mathcal{A}_{m_i} \setminus \mathcal{J}_i| \leq \frac{m_i}{2}$ and, hence, $|\mathcal{J}_i| \geq \frac{m_i}{2}$. This completes the proof. $\square$

Recall that $T_{(a,i)}$ denotes the number of times intervention $a$ is observed at context $i$. The following proposition bounds $\mathbb{E}[T_{(a,i)}]$ for each intervention $a \in \mathcal{J}_i$.

**Proposition F.2.** For every intervention $a \in \mathcal{J}_i$

$$\mathbb{E}_{\mathcal{P}_0}[T_{(a,i)}] \leq \frac{3Tr_i}{m_i}.$$

*Proof.* Any intervention $a \in \mathcal{J}_i \subseteq \mathcal{A}_{m_i}$ may be observed either when it is explicitly performed by the algorithm or as a random realization (under some other intervention, including do-nothing). Since $a \in \mathcal{A}_{m_i}$, the probability that $a$ is observed as part of some other intervention is at most $\frac{1}{m_i}$. Therefore, the expected number of times that $a$ is observed by the algorithm—without explicitly performing it—is at most $\frac{Tr_i}{m_i}$; [7] recall that the expected number of times context $i$ is visited is equal to $Tr_i$.

For any intervention $a \in \mathcal{J}_i$, by definition, the expected number of times $a$ is performed $N_{(a,i)} \leq \frac{2Tr_i}{m_i}$. Therefore, the proposition follows:

$$\mathbb{E}[T_{(a,i)}] \leq \frac{Tr_i}{m_i} + N_{(a,i)} \leq \frac{3Tr_i}{m_i}.$$

$\square$

We now state two known results for KL divergence.

**Bretagnolle-Huber Inequality (Theorem 14.2 in Lattimore & Szepesvári (2020))** : Let $\mathcal{P}$ and $\mathcal{P}'$ be any two measures on the same measurable space. Let $E$ be any event in the sample space with complement $E^c$. Then,

$$\mathbb{P}_{\mathcal{P}}\{E\} + \mathbb{P}_{\mathcal{P}'}\{E^c\} \geq \frac{1}{2} \exp\left(-\mathrm{KL}(\mathcal{P}, \mathcal{P}')\right). \tag{9}$$

**Bound on KL-Divergence with number of observations (Adaptation of Equation 17 in Lemma B1 from Auer et al. (1995))**: Let $\mathcal{P}_0$ and $\mathcal{P}_{(a,i)}$ be any two measures with differing expected rewards (for exactly the intervention $a$ at context $i$) by an amount $\beta$. Then,

$$\mathrm{KL}(\mathcal{P}_0, \mathcal{P}_{(a,i)}) \leq 6\beta^2 \, \mathbb{E}_{\mathcal{P}_0}[T_{(a,i)}] \tag{10}$$

Using this bound on KL divergence and Proposition F.2, we have, for all contexts $i \in [k]$ and interventions $a \in \mathcal{J}_i$:

$$\mathrm{KL}(\mathcal{P}_0, \mathcal{P}_{(a,i)}) \leq 6\beta^2 \cdot 3\frac{Tr_i}{m_i} = 18\frac{Tr_i\beta^2}{m_i} \tag{11}$$

---

[7] Here, we use the fact that the realization of $a$ is independent of the visitation of context $i$.

Substituting this in the Bretagnolle-Huber Inequality, we obtain, for any event $E$ in the sample space along with all contexts $i \in [k]$ and all interventions $a \in \mathcal{J}_i$:

$$\mathbb{P}_{\mathcal{P}_{(a,i)}}\{E\} + \mathbb{P}_{\mathcal{P}_0}\{E^c\} \geq \frac{1}{2} \exp\left(-18 \frac{Tr_i \beta^2}{m_i}\right) \tag{12}$$

We now define events to lower bound the probability that Algorithm $\mathcal{A}$ returns a sub-optimal policy. In particular, write $\widehat{\pi}$ to denote the policy returned by algorithm $\mathcal{A}$. Note that $\widehat{\pi}$ is a random variable.

For any $\ell \in [k]$ and any intervention $b$, write $G_1(b, \ell)$ to denote the event that—under the returned policy $\widehat{\pi}$—intervention $b$ is *not* chosen at context $\ell$, i.e., $G_1(b, \ell) := \{\widehat{\pi}(\ell) \neq b\}$. Also, let $G_2(\ell)$ denote the event that policy $\widehat{\pi}$ does not induce a transition to $\ell$ from context $0$, i.e., $G_2(\ell) := \{\widehat{\pi}(0) \neq \ell\}$. Furthermore, write $G(b, \ell) := G_1(b, \ell) \cup G_2(\ell)$. Note that the complement $G^c(b, \ell) = G_1^c(b, \ell) \cap G_2^c(\ell) = \{\widehat{\pi}(\ell) = b\} \cap \{\widehat{\pi}(0) = \ell\}$.

Considering measure $\mathcal{P}_0$, we note that for each context $\ell \in [k]$ there exists an intervention $\alpha_\ell \in \mathcal{J}_\ell$ with the property that $\mathbb{P}_{\mathcal{P}_0}\{G_1^c(\alpha_\ell, \ell)\} = \mathbb{P}_{\mathcal{P}_0}\{\widehat{\pi}(\ell) = \alpha_\ell\} \leq \frac{1}{|\mathcal{J}_\ell|}$. This follows from the fact that $\sum_{a \in \mathcal{J}_\ell} \mathbb{P}_{\mathcal{P}_0}\{\widehat{\pi}(\ell) = a\} \leq 1$. Therefore, for each context $\ell \in [k]$ there exists an intervention $\alpha_\ell$ such that $\mathbb{P}_{\mathcal{P}_0}\{G^c(\alpha_\ell, \ell)\} \leq \frac{1}{|\mathcal{J}_\ell|}$.

This bound and inequality 12 imply that for all contexts $\ell \in [k]$ there exists an intervention $\alpha_\ell$ that satisfies

$$\mathbb{P}_{\mathcal{P}_{(\alpha_\ell, \ell)}}\{G(\alpha_\ell, \ell)\} \geq \frac{1}{2} \exp\left(-18 \frac{Tr_\ell \beta^2}{m_\ell}\right) - \frac{1}{|\mathcal{J}_\ell|} \tag{13}$$

We will set

$$\beta = \min\left\{\frac{1}{3}, \sqrt{\frac{\sum_{\ell \in [k]} m_\ell}{18T}}\right\} \tag{14}$$

Therefore $\beta$ takes value either $\sqrt{\frac{\sum_{\ell \in [k]} m_\ell}{18T}}$ or $\frac{1}{3}$. We will address these over two separate cases.

**Case 1**: $\beta = \sqrt{\frac{\sum_{\ell \in [k]} m_\ell}{18T}}$.

We wish to substitute this $\beta$ value in Equation 13. Towards this, we will state a proposition.

**Proposition F.3.** There exists a context $s \in [k]$ such that

$$\sqrt{\frac{m_s}{18Tr_s}} \geq \sqrt{\frac{\sum_{\ell \in [k]} m_\ell}{18T}}$$

*Proof.* First, we note the following claim considering all vectors $\rho = \{\rho_1, \ldots, \rho_k\}$ in the probability simplex $\Delta$.

**Claim F.1.** For any given set of integers $m_1, m_2, \ldots, m_k$, we have

$$\min_{(\rho_1, \rho_2, \ldots, \rho_k) \in \Delta} \left(\max_{\ell \in [k]} \frac{m_\ell}{\rho_\ell}\right) \geq \sum_{\ell \in [k]} m_\ell$$

*Proof.* Assume, towards a contradiction, that for all $\ell \in [k]$, we have $\frac{m_\ell}{\rho_\ell} < \sum_{\ell \in [k]} m_\ell$. Then, $\rho_\ell > \frac{m_\ell}{\sum_{\ell \in [k]} m_\ell}$, for all $\ell \in [k]$. Therefore, $\sum_{\ell \in [k]} \rho_\ell > \sum_{\ell \in [k]} \frac{m_\ell}{\sum_{\ell \in [k]} m_\ell} = 1$. However, this is a contradiction as $\sum_{\ell \in [k]} \rho_\ell = 1$. □

An immediate consequence of Claim F.1 is that

$$\min_{(r_1, r_2, \ldots, r_k) \in \Delta} \left(\max_{\ell \in [k]} \sqrt{\frac{m_\ell}{18Tr_\ell}}\right) \geq \sqrt{\frac{\sum_{\ell \in [k]} m_\ell}{18T}}$$

.

Therefore, irrespective of how $r_i$s are chosen, there always exists a context $s \in [k]$ such that
$\sqrt{\frac{m_s}{18Tr_s}} \geq \sqrt{\frac{\sum_{\ell \in [k]} m_\ell}{18T}}$. □

For such a context $s \in [k]$ that satisfies Proposition F.3, we note that, $\frac{m_s}{18Tr_s} \geq \beta^2$ or $\frac{18Tr_s\beta^2}{m_s} \leq 1$.

Let us now restate Equation 13 for such a context $s$. There exists a context $s \in [k]$ and an intervention $\alpha_s$ that satisfies

$$\mathbb{P}_{\mathcal{P}_{(\alpha_s,s)}}\{G(\alpha_s, s)\} \geq \frac{1}{2}\exp\left(-18\frac{Tr_s\beta^2}{m_s}\right) - \frac{1}{|\mathcal{J}_s|} \geq \frac{1}{2e} - \frac{1}{|\mathcal{J}_s|} \tag{15}$$

Note that the last inequality lower bounds the to probability of selecting a non-optimal policy when the algorithm $\mathcal{A}$ is executed on instance $\mathcal{F}_{\alpha_s,s}$. Furthermore, in instance $\mathcal{F}_{\alpha_s,s}$, for any non-optimal policy $\widehat{\pi}$ we have $\varepsilon(\widehat{\pi}) = \left(\frac{1}{2} + \beta\right) - \frac{1}{2} = \beta$. Therefore, we can lower bound $\mathcal{A}$'s regret over instance $\mathcal{F}_{\alpha_s,s}$ as follows:

$$\text{Regret}_T = \mathbb{E}[\varepsilon(\widehat{\pi})] = \mathbb{P}_{\mathcal{P}_{(\alpha_s,s)}}\{G(\alpha_s, s)\} \cdot \mathbb{E}[\text{Regret} \mid G(\alpha_s, s)] + \tag{16}$$
$$\mathbb{P}_{\mathcal{P}_{(\alpha_s,s)}}\{G^c(\alpha_s, s)\} \cdot \mathbb{E}[\text{Regret} \mid G^c(\alpha_s, s)]$$
$$\geq \left[\frac{1}{2e} - \frac{1}{|\mathcal{J}_s|}\right]\beta + \mathbb{P}_{\mathcal{P}_{(\alpha_s,s)}}\{G^c(\alpha_s, s)\} \cdot 0$$
$$= \left[\frac{1}{2e} - \frac{1}{|\mathcal{J}_s|}\right]\beta \tag{17}$$

Note that we can construct the instances to ensure that $m_\ell \geq 8$, for all contexts $\ell$, and, hence, $\left(\frac{1}{2e} - \frac{1}{|\mathcal{J}_i|}\right) = \Omega(1)$ (see Proposition F.1). Therefore Equation 17 gives us:

$$\text{Regret}_T = \Omega(\beta) = \Omega\left(\sqrt{\frac{\sum_{\ell \in [k]} m_\ell}{T}}\right) \tag{18}$$

**Case 2** We now consider the case when $\beta = \frac{1}{3}$. In such a case, $\sqrt{\frac{\sum_{\ell \in [k]} m_\ell}{18T}} > \frac{1}{3}$.

We showed in Proposition F.3 that there exists a context $s \in [k]$ such that $\sqrt{\frac{m_s}{18Tr_s}} \geq \sqrt{\frac{\sum_{\ell \in [k]} m_\ell}{18T}}$. Combining the two statements, there exists a context $s$ such that $\sqrt{\frac{m_s}{18Tr_s}} \geq \frac{1}{3}$. We now restate Inequality 13 for such a context $s \in [k]$:

$$\mathbb{P}_{\mathcal{P}_{(\alpha_s,s)}}\{G(\alpha_s, s)\} \geq \frac{1}{2}\exp\left(-9\beta^2\right) - \frac{1}{|\mathcal{J}_s|} = \frac{1}{2e} - \frac{1}{|\mathcal{J}_s|}$$

Following the exact same procedure as in Case 1, we can derive that $\text{Regret}_T \geq \left[\frac{1}{2e} - \frac{1}{|\mathcal{J}_s|}\right]\beta$.

We saw in Case 1 that it is possible to construct instances such that $\left[\frac{1}{2e} - \frac{1}{|\mathcal{J}_s|}\right] = \Omega(1)$. Therefore the following holds for Case 2 also:

$$\text{Regret}_T = \Omega(\beta) = \Omega\left(\sqrt{\frac{\sum_{\ell \in [k]} m_\ell}{T}}\right) \tag{19}$$

Inequalities 18 and 19 imply that there exists a context $s$ and an intervention $\alpha_s$ such that, under instance $\mathcal{F}_{(\alpha_s,s)}$, algorithm $\mathcal{A}$'s regret satisfies

$$\text{Regret}_T = \Omega\left(\sqrt{\frac{\sum_{\ell \in [k]} m_\ell}{T}}\right) \tag{20}$$

We complete the proof of Theorem 2 by showing that in the current context $\lambda = \sum_{\ell \in [k]} m_\ell$.

**Proposition F.4.** For the chosen transition matrix

$$\lambda := \min_{\text{fq. vector} f} \left\| PM^{1/2} \left( P^\top f \right)^{\circ - \frac{1}{2}} \right\|_\infty^2 = \sum_{\ell \in [k]} m_\ell$$

*Proof.* Recall that all the instances, $\mathcal{F}_0$ and $\mathcal{F}_{(a,i)}$s, have the same (deterministic) transition matrix $P$. Also, parameter $\lambda$ is computed via Equation 3.

Consider any frequency vector $f$ over the interventions $\{1, \ldots, N\}$. From the chosen transition matrix, we have the following:

$$
P = \begin{bmatrix} 1 & 0 & \ldots & 0 \\ 0 & 1 & \ldots & 0 \\ & & \ldots & \\ 0 & 0 & \ldots & 1 \\ & & \ldots & \\ 0 & 0 & \ldots & 1 \end{bmatrix} \quad
PM^{\frac{1}{2}} = \begin{bmatrix} \sqrt{m_1} & 0 & \ldots & 0 \\ 0 & \sqrt{m_2} & \ldots & 0 \\ & & \ldots & \\ 0 & 0 & \ldots & \sqrt{m_k} \\ & & \ldots & \\ 0 & 0 & \ldots & \sqrt{m_k} \end{bmatrix} \quad
P^\top f = \begin{bmatrix} f_1 \\ f_2 \\ \ldots \\ f_{k-1} \\ f_k + \ldots + f_N \end{bmatrix}
$$

From here, we can compute the following:

$$PM^{1/2} \left( P^\top f \right)^{\circ - \frac{1}{2}} = \left[ \sqrt{\frac{m_1}{f_1}}, \ldots, \sqrt{\frac{m_{k-1}}{f_{k-1}}}, \sqrt{\frac{m_k}{f_k + \ldots + f_N}}, \ldots, \sqrt{\frac{m_k}{f_k + \ldots + f_N}} \right]^\top$$

That is, for all $\ell \in [k-1]$, the $\ell$th component of the vector $PM^{1/2} \left( P^\top f \right)^{\circ - \frac{1}{2}}$ is equal to $\sqrt{\frac{m_i}{f_i}}$. All the remaining components are $\sqrt{\frac{m_k}{f_k + \ldots + f_N}}$.

Write $\rho_\ell := f_\ell$ for all $\ell \in [k-1]$ and $\rho_k = \sum_{j=k}^{N} f_j$. Since $f$ is a frequency vector, $(\rho_1, \ldots \rho_k) \in \Delta$. In addition,

$$PM^{1/2} \left( P^\top f \right)^{\circ - \frac{1}{2}} = \left[ \sqrt{\frac{m_1}{\rho_1}}, \ldots, \sqrt{\frac{m_{k-1}}{\rho_{k-1}}}, \sqrt{\frac{m_k}{\rho_k}}, \ldots, \sqrt{\frac{m_k}{\rho_k}} \right]^\top$$

Therefore, by definition, $\lambda = \min_{(\rho_1, \ldots, \rho_k) \in \Delta} \left( \max_{\ell \in [k]} \frac{m_\ell}{\rho_\ell} \right)$. Now, using a complementary form of Claim F.1 we obtain $\lambda = \sum_{\ell \in [k]} m_\ell$. The proposition stands proved.

$\square$

Finally, substituting Proposition F.4 into Equation 20, we obtain that there exists an instance $\mathcal{F}_{(\alpha_s, s)}$ for which algorithm $\mathcal{A}$'s regret is lower bounded as follows

$$\text{Regret}_T = \Omega \left( \sqrt{\frac{\lambda}{T}} \right). \tag{21}$$

This completes the proof of Theorem 2.

F.2 PROOF OF INEQUALITY (10)

For completeness, we provide a proof of inequality (10).

**Lemma 12.** $\text{KL}(\mathcal{P}_0, \mathcal{P}_{(a,i)}) \le 6\beta_i^2 \, \mathbb{E}_{\mathcal{P}_0}[\text{T}_{(a,i)}]$

*Proof of Inequality (10).* This proof is based on lemma B1 in Auer et al. (1995). We define a couple of notations for this proof. Let $\mathbf{R}_{t-1}$ indicate the filtration (of rewards and other observations) up to time $t-1$. and $R_t$ indicate the reward at time $t$ for this proof.

$$\text{KL}(\mathcal{P}_0, \mathcal{P}_{(a,i)}) = \text{KL} \left[ \mathbb{P}_{\mathcal{P}_0}(\text{R}_T, \text{R}_{T-1}, \ldots, \text{R}_1) \| \mathbb{P}_{\mathcal{P}_{(a,i)}}(\text{R}_T, \text{R}_{T-1}, \ldots, \text{R}_1) \right]$$

We now state (without proof) a useful lemma for bounding the KL divergence between random variables over a number of observations.

**Chain Rule for entropy (Theorem 2.5.1 in Cover & Thomas (2006)):** Let $X_1, \ldots, X_T$ be random variables drawn according to $P_1, \ldots, P_T$. Then

$$H(X_1, X_2, \ldots, X_T) = \sum_{i=1}^{T} H(X_i \mid X_{i-1}, X_{i-2}, \ldots, X_1)$$

where $H(\cdot)$ is the entropy associated with the random variables.

Using the chain rule for entropy

$$\mathrm{KL}(\mathcal{P}_0, \mathcal{P}_{(a,i)}) = \sum_{t=1}^{T} \mathrm{KL}\left[ \mathbb{P}_{\mathcal{P}_0}(\mathrm{R}_t \mid \mathbf{R}_{t-1}) \parallel \mathbb{P}_{\mathcal{P}_{(a,i)}}(\mathrm{R}_t \mid \mathbf{R}_{t-1}) \right]$$

Let $a_t$ be the intervention chosen by the Algorithm $\mathcal{A}$ at time $t$. Then:

$$= \sum_{t=1}^{T} \mathbb{P}_{\mathcal{P}_0}\{a_t \neq a \mid \mathbf{R}_{t-1}\} \left( \frac{1}{2} \parallel \frac{1}{2} \right) + \mathbb{P}_{\mathcal{P}_0}\{a_t = a \mid \mathbf{R}_{t-1}\} \mathrm{KL}\left( \frac{1}{2} \parallel \frac{1}{2} + \beta_i \right)$$

Since $\mathrm{KL}\left( \frac{1}{2} \parallel \frac{1}{2} \right) = 0$, we get:

$$= \sum_{t=1}^{T} \mathbb{P}_{\mathcal{P}_0}\{a_t = a \mid \mathbf{R}_{t-1}\} \mathrm{KL}\left( \frac{1}{2} \parallel \frac{1}{2} + \beta_i \right)$$

$$= \mathrm{KL}\left( \frac{1}{2} \parallel \frac{1}{2} + \beta_i \right) \sum_{t=1}^{T} \mathbb{P}_{\mathcal{P}_0}\{a_t = a \mid \mathbf{R}_{t-1}\}$$

$$= \mathrm{KL}\left( \frac{1}{2} \parallel \frac{1}{2} + \beta_i \right) \mathbb{E}_{\mathcal{P}_0}[\mathrm{T}_{(a,i)}]$$

**Claim F.2.** $\mathrm{KL}\left( \frac{1}{2} \parallel \frac{1}{2} + \beta_i \right) = -\frac{1}{2}\log_2(1 - 4\beta_i^2) \leq 6\beta_i^2$

*Proof.*

$$\mathrm{KL}\left( \frac{1}{2} \parallel \frac{1}{2} + \beta_i \right) = \frac{1}{2}\log_2\left[ \frac{\frac{1}{2}}{\frac{1}{2} + \beta_i} \right] + (1 - \frac{1}{2})\log_2\left[ \frac{(1 - \frac{1}{2})}{(1 - \frac{1}{2} - \beta_i)} \right]$$

$$= \frac{1}{2}\log_2\left[ \frac{1}{1 + 2\beta_i} \right] + \frac{1}{2}\log_2\left[ \frac{1}{1 - 2\beta_i} \right]$$

$$= \frac{1}{2}\log_2\left[ \frac{1}{1 - 4\beta_i^2} \right] = -\frac{1}{2}\log_2\left[ 1 - 4\beta_i^2 \right]$$

$$= -\frac{1}{2\ln(2)}\ln\left[ 1 - 4\beta_i^2 \right] \leq \frac{4\beta_i^2}{2\ln(2)} < 6\beta_i^2$$

where the last inequality is obtained from the Taylor series expansion of the log. $\square$

It follows that: $\mathrm{KL}(\mathbb{P}_0, \mathbb{P}_1) \leq 6\beta_i^2 \mathbb{E}_{\mathcal{P}_0}[\mathrm{T}_{(a,i)}]$. $\square$