# OpenReview forum: "Learning Good Interventions in Causal Contextual Bandits with Adaptive Context"
_ICLR.cc/2024/Conference — Submitted to ICLR 2024_

### Official Review · Reviewer_fZFX · 2023-10-29

**Soundness:** 3 good
**Presentation:** 2 fair
**Contribution:** 2 fair
**Rating:** 5
**Confidence:** 4

**Summary:**

This paper considers a new type of causal bandit problem, where the decision maker makes two decisions, receives one context in between and a final reward. A motivating online advisement example is provided. An efficient exploration algorithm called ConvExplore is provided and its corresponding regret is shown, which serves as an upper bound for the proposed problem setting. In addition, a lower bound result is provided, which indicates that the proposed algorithm is tight up to log factors.

**Strengths:**

- The problem setting is new, which allows the context distribution (graph) to be stochastically depending on the initial action, e.g., user type selection.
- The proposed algorithm is clear, which consists three subroutines: estimating transition probabilities, causal parameters and the corresponding rewards.

**Weaknesses:**

- If I understood this paper correctly, all k contexts are not correlated. The contexts represent different users, and each user can have totally different causal graph and reward function. The proposed algorithm is learning the probabilities and reward functions independently among all contexts. In practice, the number of context could be very large, the proposed algorithm is not very practical as it requires a lot of explorations.
- The causal graph setting in this paper is not new and is very rudimental, i.e., the assumption of the graph for each context, which is similar to the original Lattimore 2016 paper and ignores many recent developments, e.g., Lu 2020, 2021, Adaptively Exploiting d-Separators with Causal Bandits 2022.

**Questions:**

- Is it possible to share some knowledge between different context graphs and/or reward functions? Is it possible to define the contexts as different user groups that share similar behaviors in terms of features and rewards?
- The word context is bit confusing here, especially in Figure 2, the context in contextual bandits and in this paper's setting are quite different.
- In terms of the analysis, how different is it compared to Subramanian 2022 work, e.g., can you explain why the stochastic transition to the contexts will make the analysis in the paper much more challenging?

---

> ### Author Response · Authors · 2023-11-23
> **Response to Reviewer fZFX**
>
> We thank the reviewer for appreciating the novelty of our problem setting and for further highlighting the clarity of the Algorithm. We address some of the reviewer's questions below:
>
> 1. The k contexts are not correlated. The contexts represent different users, and each user can have a totally different causal graph and reward function.
>
> A1. In our manuscript, the term 'context' is used in a specific manner, which might not have been evidently clear. It does not refer to a single user, but rather to their demographic. Each demographic may have many users, and as such, the number of contexts is driven by the amount of information an advertiser receives about a user from the platform (which is limited). In practice, each context -- of which there are a finite number -- may contain a large number of users on the platform. Further we note that having correlated contexts may require additional assumptions as to how information is shared between contexts. Once such assumptions are incorporated, our framework is general enough to accomodate such a problem setup. Having said that, correlated contexts is not the problem we are addressing. We are happy to provide additional information or elaboration on this if it would be helpful.
>
> 2. The causal graph setting in this paper is not new.
>
> A2. While newer causal graph settings have been considered in Bilodeau et al. (2022) and Lu et al. (2021), our contributions are altogether novel with regard to the decision problem considered. Having said that, we thank the reviewer for pointing us to these references. We will add, “There have been new lines of work extending the causal graph considered in this setting in Lu et al. (2021), alongside works that consider a d-separation between the intervention and the outcome (Bilodeau et al., 2022)."
>
> 3. Why are stochastic transitions to the contexts difficult?
>
> A3. We address this question in three parts: the difficulty in practice, the difficulty in the setting, and the difficulty in analysis.
>
> First, we address the practical difficulty. We note that in practice, it is not possible to request a platform for exactly one type of user that we want. For example, if we request a demographic of new-parents, we may be referred to users who are either high-income or low-income (who both share the attribute of being a new parent). There are various other attributes of users that may or may not be requested and may be assigned to us. Therefore, assuming a deterministic transition to a context is practically infeasible.
>
> Second, we address the difficulty in the setting. To compare the two settings, we carried out various experiments. We share the results below:
>    1. Plot for regret with lambda in deterministic and nondeterministic settings:
>       [Link to Plot](https://drive.google.com/file/d/1l3_jhuOS3xz_8LcQAaZhTbDfvNJxktHL/view?usp=sharing)
>
>    2. Plot for regret with exploration budget in deterministic and nondeterministic settings:
>       [Link to Plot](https://drive.google.com/file/d/1qi0ns6ZCP4RZP3RCJ8MymDafvGZRFfG3/view?usp=sharing)
>
>    3. Plot for regret with the number of intermediate contexts:
>       [Link to Plot](https://drive.google.com/file/d/1pJ-Ya3Fhclb0fowOWRY0nS_nb3Gk4y8g/view?usp=sharing)
>
> We note that in each of these cases, the stochastic cases are harder.
>
> We provide plots for several other experiments based on reviewer suggestions below, which we will be adding to the manuscript:
> [Additional Experiments](https://drive.google.com/drive/folders/1VMkeenDM797NtsR25_Fnsc3t3yZkuqy1?usp=sharing)
>
> Our anonymous GitHub repo contains the code required to replicate these experiments:
> [causal contextual bandit GitHub repo](https://github.com/adaptiveContextualCausalBandits/aCCB)
>
> Third, we address the difficulty in the analysis of the stochastic problem. Consider only two interventions at the start state, separated by an $\epsilon$ probability for the transition to a preferred context that has an optimal reward. Separating these two interventions at the start state becomes extremely difficult. The deterministic transitions entirely sidestep these concerns and can directly reach the preferred context. Therefore, the analysis in such a stochastic setting is hard.
>
> ---
>
> We hope the above points clarify the reviewer’s concerns. We are happy to expand on our manuscript based on the reviewer's feedback. Should you find that these revisions meet your expectations, we would kindly request you to consider revising your evaluation score accordingly. We appreciate your time and effort in reviewing our work.

---

### Official Review · Reviewer_oUko · 2023-10-30

**Soundness:** 3 good
**Presentation:** 2 fair
**Contribution:** 2 fair
**Rating:** 5
**Confidence:** 3

**Summary:**

The paper studies a variant of causal contextual bandits where the learner's initial action influences the context. The objective is to identify optimal interventions in the initial state and post-context identification. The study extends previous work from deterministic to stochastic contexts and offers a regret minimization guarantee using a parameter called λ. The research demonstrates that these guarantees are tight for a broad range of instances. Notably, the work employs convex optimization to address the bandit exploration problem and includes experimental validation of the theoretical results.

**Strengths:**

(1) The idea of using the convex minimization problem $\lambda$ is interesting.
(2) Provide both upper bound and lower bound for the proposed algorithm.

**Weaknesses:**

(1) Authors should compare the convex exploration with other exploration strategies, such as UCB-based or TS-based, instead of only uniform exploration.
(2) The authors didn't provide a comparison of their regret bound with other related works, given that there are plenty of causal bandit works.
(3) What is the upper bound or lower bound of $\lambda$?
(4) The overall writing looks like finishing in a rush.

**Questions:**

See weakness.

---

> ### Author Response · Authors · 2023-11-23
> **Response to Reviewer oUko**
>
> We thank the reviewer for appreciating our reduction of the bandit problem to a convex minimization problem. We address some of the reviewer's questions below:
>
>
> 1. Authors should compare the convex exploration with other exploration strategies, such as UCB-based or TS-based, instead of only uniform exploration.
> A1. We thank the reviewer for these thoughtful comments. To compare our algorithm with these baselines, we carried out several new experiments, the results of which we share below.
>
> # Variation of Expected Regret with Exploration Budget
>
> | exploration_budget | UE				 	 | roundrobin_ucb | roundrobin_ts | ucb_over_intervention_pairs | ts_over_intervention_pairs | convex_explorer |
> |--------------------|-----------------------|----------------|---------------|-----------------------------|----------------------------|-----------------|
> | 100                | 1.0                   | 1.0            | 1.0           | 1.0                         | 1.0                        | 0.87            |
> | 250                | 0.99                  | 0.95           | 0.96          | 1.0                         | 1.0                        | 0.44            |
> | 500                | 0.71                  | 0.54           | 0.97          | 0.99                        | 1.0                        | 0.16            |
> | 1,000              | 0.4                   | 0.32           | 0.96          | 0.79                        | 1.0                        | 0.02            |
> | 2,500              | 0.02                  | 0.1            | 0.96          | 0.2                         | 1.0                        | 0.0             |
> | 5,000              | 0.0                   | 0.04           | 0.94          | 0.13                        | 1.0                        | 0.0             |
> | 7,500              | 0.0                   | 0.01           | 0.96          | 0.1                         | 1.0                        | 0.0             |
> | 10,000             | 0.0                   | 0.0            | 0.93          | 0.08                        | 1.0                        | 0.0             |
> | 12,500             | 0.0                   | 0.0            | 0.91          | 0.06                        | 1.0                        | 0.0             |
> | 15,000             | 0.0                   | 0.0            | 0.93          | 0.06                        | 1.0                        | 0.0             |
> | 20,000             | 0.0                   | 0.0            | 0.94          | 0.0                         | 1.0                        | 0.0             |
> | 25,000             | 0.0                   | 0.0            | 0.96          | 0.0                         | 1.0                        | 0.0             |
> | 100,000            | 0.0                   | 0.0            | 0.97          | 0.0                         | 1.0                        | 0.0             |
>
>
> Firstly, we note that the uniform exploration algorithm (round robin) performs the best amongst the non-causal algorithms. UCB for choice of intervention at each context, with round-robin over the choice of initial intervention also works quite well, but takes a longer time to converge to the right solution. Thompson sampling based approaches do not work well. The works performing algorithms were those which treated the intervention pairs as a bandit, and tried to optimize over this larger bandit problem (ignoring the information available in the structure of the problem). We share the plot related to this table below:
> https://drive.google.com/file/d/1qWSt7Kv-sEi85dD4sjflLnQqRC7V_TCN/view?usp=sharing (Note the log scaling over the x-axis).
>
> We provide plots for several other experiments based on reviewer suggestions below, which we will be adding to the manuscipt:
> https://drive.google.com/drive/folders/1VMkeenDM797NtsR25_Fnsc3t3yZkuqy1?usp=sharing
>
> Our anonymous github repo contains the code required to replicate these experiments:
> https://github.com/adaptiveContextualCausalBandits/aCCB
>
>
> 2. The authors didn't provide a comparison of their regret bound with other related works
> A2. We have added several new baselines to our experimental work presently (cf. our [causal contextual bandit GitHub repo](https://github.com/adaptiveContextualCausalBandits/aCCB)). Many other works are not directly applicable to our model. Due to our setting extending those considered in prior works, we note that they may not be directly applicable here. For example, causal bandit, or causal contextual bandit works are not applicable to causal contextual bandit works with adaptive contexts.
>
> Having said that, we thank the reviewer for bringing up this point. We will add a paragraph explaining (a) baselines we have compared with and (b) why some of the baselines from other causal bandit works are not applicable.

---

> ### Author Response · Authors · 2023-11-23
> **Response to Reviewer oUko**
>
> 3. What is the upper bound or lower bound of $\lambda$?
>
> A3. The lower bound on $\lambda$ is $2k$ and upperbound is $nk$. While we have noted the upperbound in page 7 ("The upperbound on $\lambda$ is $nk$") and other places, we will clarify on the lower bound as well.
>
> ---
>
> We indeed thank the reviwer for their suggestions on our experimental section. We were able to expand on this section greatly due to the suggestion to compare with baselines such as TS and UCB.
>
> We hope that the responses provided above have adequately addressed the reviewer's concerns. Should the reviewer find that these revisions meet their expectations, we would kindly request them to consider revising their evaluation score.
>
> We deeply appreciate the reviewer's time and effort in reviewing our work.

---

### Official Review · Reviewer_73yj · 2023-11-02

**Soundness:** 2 fair
**Presentation:** 2 fair
**Contribution:** 2 fair
**Rating:** 5
**Confidence:** 3

**Summary:**

This paper proposed a new method called ConvExplore(CE) to solve the Causal Contextual Bandits(CCBs). To be specific, it handle the CCBs with adaptive context settings, which uses an instance-dependent causal parameter \lambda to make adaptions to different contexts. Authors also provided solid proofs and regret bound of their new method and made experiments to validate their theoretical results.

**Strengths:**

1. Solid proofs of minimizing simple regret for causal bandits with adaptive context in an intervention efficient manner.
2. Upper and lower bound of the simple regret acheived by CE indicates that authors' method is almost the ideal solution for CCBs.

**Weaknesses:**

1.From the experimental results, it is observed that the choice of λ significantly influences the algorithm's performance comparison. Could authors provide the variation curve under larger λ values?

2.Moreover, how should λ be specifically adjusted, especially when dealing with entirely new contexts in a new scenario? Additionally, when λ is less than nk (e.g., λ=390), CE's performance is inferior to that of UE. How can this be explained?

3.There are too few experimental results, and is it possible to provide the source or generation rules of the experimental data? Although the theoretical aspects are solid, the experimental section needs further improvement, especially in terms of interpretability. The motivation given earlier pertains to a cold start scenario. Could authors provide further explanations for the experiments?

**Questions:**

See Questions in Weaknesses.

---

> ### Author Response · Authors · 2023-11-23
> **Response to Reviewer 73yj**
>
> We thank the reviewer for several positive comments made on the paper, especially onthe strength of our proof and for indicating that our method is highly suitable for CCBs.
>
>
> We address the points raised in the 'Weaknesses' and 'Questions' sections below.
> 1. 1.From the experimental results, it is observed that the choice of λ significantly influences the algorithm's performance comparison. Could authors provide the variation curve under larger λ values?
> A1. In fact, we note that we have varied lambda up to the largest possible value that it can take theoretically. At the right end, the lambda has been taken to be $nk$, which is the largest value it can possibly take.
>
> 2. How should $\lambda$ be specifically adjusted
> A2 The $\lambda$ value is based on the instance given to us, and is not an adjustable hyperparameter. It is something intrinsic to the hardness of the problem given to us. In most instances, $\lambda$ is likely to be low given causal variables do not have $q$ values which are close to 0.
>
> 2b. CE's performance is inferior to that of UE. How can this be explained?
> A2b. In the theoretical analysis, we were not utilising the data from the first two parts of the algorithm (where the probability of transition, and causal parameter were being estimated). This is because these parts of the algorithm were only influencing the constants in the regret term. But, in a practical implementation, the data from these parts of the algorithm can also be used to estimate rewards. We have done so now, and the following are the updated results (with more baselines as well):
> https://drive.google.com/file/d/1YydC2e_NkCBWEp-IAPB0EXNPPzvwt3mU/view?usp=sharing
>
> Further, we provide the data related to the figure here:
>
> # Variation of Expected Regret with Lambda $\lambda$
>
> | lambda    | UE			 		| roundrobin_ucb | roundrobin_ts | ucb_over_intervention_pairs | ts_over_intervention_pairs | convex_explorer |
> |-----------|-----------------------|----------------|---------------|-----------------------------|----------------------------|-----------------|
> | 20.000000 | 0.152000              | 0.327999       | 1.000005      | 0.395999                    | 1.000005                   | 0.000000        |
> | 30.000000 | 0.196000              | 0.321999       | 1.000005      | 0.337999                    | 1.000005                   | 0.000000        |
> | 40.000000 | 0.164000              | 0.333999       | 1.000005      | 0.391999                    | 1.000005                   | 0.002000        |
> | 49.999996 | 0.140000              | 0.312000       | 1.000005      | 0.391999                    | 1.000005                   | 0.000000        |
> | 60.000000 | 0.182000              | 0.339999       | 1.000005      | 0.361999                    | 1.000005                   | 0.002000        |
> | 70.000000 | 0.180000              | 0.312000       | 1.000005      | 0.365999                    | 1.000005                   | 0.018000        |
> | 79.999990 | 0.176000              | 0.310000       | 1.000005      | 0.349999                    | 1.000005                   | 0.038000        |
> | 90.000000 | 0.192000              | 0.290000       | 1.000005      | 0.381999                    | 1.000005                   | 0.060000        |
> | 100.000000| 0.174000              | 0.353999       | 1.000005      | 0.385999                    | 1.000005                   | 0.082000        |
>
> We hope this clarifies that CE is not worse the UE (round robin) across domains.
>
> 3. The experimental section needs further improvement
> A3. We thank the reviewer for their well thought our suggestion in this point. We have carried out eight further experiments to this work currently, including adding more baselines and plotting the regret across various instances.
>
> We share the plots related to our new experiments here:
> https://drive.google.com/drive/folders/1VMkeenDM797NtsR25_Fnsc3t3yZkuqy1
>
> Our anonymous github repo contains the code required to replicate these experiments:
> https://github.com/adaptiveContextualCausalBandits/aCCB
>
> 3b. Is it possible to provide the source or generation rules
> A3 To share our generation rules and provide further code-replicability, we share our entire code on our [causal contextual bandit GitHub repo](https://github.com/adaptiveContextualCausalBandits/aCCB). We will further add a full section expanding on our experimental setup in the supplementary material.
>
>
>
> We hope that the responses provided above have adequately addressed your concerns. We are grateful for your insightful comments, which have been instrumental in improving our manuscript. Should you find that these revisions meet your expectations, we would kindly request you to consider revising your evaluation score accordingly. We appreciate your time and effort in reviewing our work.

---

### Official Review · Reviewer_aW5H · 2023-11-22

**Soundness:** 3 good
**Presentation:** 2 fair
**Contribution:** 2 fair
**Rating:** 5
**Confidence:** 3

**Summary:**

This paper studies the causal contextual bandits, where the context depends on the learner's initial action. The goal is to select the actions (causal interventions) before and after observing the context that maximizes the reward. To achieve this, the authors propose an algorithm with better simple regret.

**Strengths:**

**The following are the key strengths of the paper:**
1. This paper studies how causal structure can improve contextual bandit algorithms where context depends on the initial action taken by the learner. This problem has real-life applications in areas like online advertisement (as mentioned in the paper).

2. The authors propose an algorithm (ConvExplore) for the problem considered in the paper and show that it enjoys better simple regret, and empirical results also verify the theoretical results.

**Weaknesses:**

**The following are the key weaknesses of the paper:**
1. Restricting to binary interventions and rewards (i.e., either 0 or 1) makes the problem easier to solve but limits the practical applications to problems with only binary interventions and rewards.

2. The possible number of contexts can be very large (or even infinite), e.g., the number of users on the platform. Therefore, working with matrix P (where the number of columns is the same as the number of contexts) may be computationally challenging.

3. The proposed algorithm is horizon-dependent as it needs to know the total number of rounds, T, upfront. It is unclear if the proposed algorithm will work for problems where T is very small.

**Questions:**

Please address the above weaknesses. I have a few more questions/comments:
1. Page 1, last paragraph: what are the other variables in this statement, "they get to observe the values of multiple other variables in the causal graph."?

2. Is there any relationship between $\lambda$ and effective dimension if the problem is modeled as a sparse bandit problem?

---

> ### Author Response · Authors · 2023-11-23
> **Response to Reviewer aW5H**
>
> We thank the reviewer for several positive comments made on the paper.
>
>
> We address the points raised in the 'Weaknesses' and 'Questions' sections below.
>
> 1. Restricting to binary interventions and rewards (i.e., either 0 or 1) makes the problem easier to solve but limits the practical applications to problems with only binary interventions and rewards.
>
> A1. The binary intervention setting is not really restrictive  as the setting is general enough to capture all finite alphabets (over the variables). We will clarify this point further in the manuscript.
>
> 2. The possible number of contexts can be very large (or even infinite), e.g., the number of users on the platform. Therefore, working with matrix P (where the number of columns is the same as the number of contexts) may be computationally challenging.
> A2. A contextual bandit setting need not hold one context per unit or per user in the system. Often, one may find many users pertain to the same context. For example, if one assume a combination of user-device and user-demographic to be a context, then not only is the number of contexts small, but further, the number of users per context may also be large. Crucially, irrespective of the number of contexts, we perform better than alternatives algorithms for this setting where regret is upper bounded by $\sqrt{nk/T}$ for n contexts and k interventions per context. Our regret is upper bounded by $\sqrt{\lambda/T}$ where $\lambda \ll nk$.
>
> 3. The proposed algorithm is horizon-dependent as it needs to know the total number of rounds, T, upfront.
> A3. Yes indeed. This is a standard assumption in the simple regret setting. These settings are applicable in domains where there is some budget given for exploring, in which time the near-optimal solution has to be found. For example, in medical domains, a certain number of exploratory tests are allowed after which the best variant of a drug has to be found. In the advertising domain, marketers often look at A/B tests where there is a certain budget for exploration after the marketer makes their decision on the best arm/intervention.
>
> 4. Page 1, last paragraph: what are the other variables in this statement, "they get to observe the values of multiple other variables in the causal graph."?
> A4. Since variables in a causal graph are causally linked as in a causal bayesian network (CBN), by intervening on one variable in the graph, the values of other variables may be inferred. Specifically, by doing the do-nothing intervention, one may observe the values of multiple variables in the causal graph as well as the reward variable. Thereby a causal intervention is more potent than an arm-pull in a multi-armed bandit setting.
>
> 5. Is there any relationship between $\lambda$ and effective dimension if the problem is modeled as a sparse bandit problem?
> A5. Indeed there is! As the reviewer has observed the causal parameter $\lambda$ is dependent on various features of the problem, including the number of contexts, and the causal threshold parameter $m$ at each context. Having said that, in our understanding, the modelling of the problem as a sparse bandit may not help.
>
> To experimentally verify this, we carried out an experiment where we compare our model (convex_explorer) with two baseline models (ucb_over_intervention_pairs,ts_over_intervention_pairs) which treat the problem as a large bandit problem. These models treat the pair of interventions at the start state, and at the intermediate context -- which may be thought of as the (user_demographic, advertisement_choice) pair -- as a large bandit problem. We note that such a formulation leads to sub-optimal regret as seen in the figure here: https://drive.google.com/file/d/1YydC2e_NkCBWEp-IAPB0EXNPPzvwt3mU/view?usp=sharing
>
> We propose that such sub-optimal regret is because the information regarding the context is not shared, and hence the regret is high.

---

> > ### Author Response · Authors · 2023-11-23
> > **Data used to generate the figure**
> >
> > The manuscript will be updated to reflect the experiment in the figure:
> > https://drive.google.com/file/d/1YydC2e_NkCBWEp-IAPB0EXNPPzvwt3mU/view?usp=sharing
> >
> > Our anonymous github repo contains the code required to replicate this experiment:
> > https://github.com/adaptiveContextualCausalBandits/aCCB
> >
> >
> > Further, we provide the data related to the figure here:
> >
> > # Variation of Expected Regret with Lambda $\lambda$
> >
> > | lambda    | roundrobin_roundrobin | roundrobin_ucb | roundrobin_ts | ucb_over_intervention_pairs | ts_over_intervention_pairs | convex_explorer |
> > |-----------|-----------------------|----------------|---------------|-----------------------------|----------------------------|-----------------|
> > | 20.000000 | 0.152000              | 0.327999       | 1.000005      | 0.395999                    | 1.000005                   | 0.000000        |
> > | 30.000000 | 0.196000              | 0.321999       | 1.000005      | 0.337999                    | 1.000005                   | 0.000000        |
> > | 40.000000 | 0.164000              | 0.333999       | 1.000005      | 0.391999                    | 1.000005                   | 0.002000        |
> > | 49.999996 | 0.140000              | 0.312000       | 1.000005      | 0.391999                    | 1.000005                   | 0.000000        |
> > | 60.000000 | 0.182000              | 0.339999       | 1.000005      | 0.361999                    | 1.000005                   | 0.002000        |
> > | 70.000000 | 0.180000              | 0.312000       | 1.000005      | 0.365999                    | 1.000005                   | 0.018000        |
> > | 79.999990 | 0.176000              | 0.310000       | 1.000005      | 0.349999                    | 1.000005                   | 0.038000        |
> > | 90.000000 | 0.192000              | 0.290000       | 1.000005      | 0.381999                    | 1.000005                   | 0.060000        |
> > | 100.000000| 0.174000              | 0.353999       | 1.000005      | 0.385999                    | 1.000005                   | 0.082000        |
> >
> > Here, the baselines 'ucb_over_intervention_pairs' and 'ts_over_intervention_pairs' treat the problem as a large bandit instance, with the action space being pairs of interventions.
> >
> > Experimental plots based on our updated experimental work:
> > https://drive.google.com/drive/folders/1VMkeenDM797NtsR25_Fnsc3t3yZkuqy1
> >
> >
> >
> > We hope the above points clarify the reviewer’s concerns. We are happy to expand on our manuscript based on the reviewer's feedback. Should you find that these revisions meet your expectations, we would kindly request you to consider revising your evaluation score accordingly. We appreciate your time and effort in reviewing our work.

---

### Author Response · Authors · 2023-11-23
**We thank reviewers for their feedback**

Dear reviewers,

Thank you for taking the time so far to offer us constructive feedback on our paper. In some responses we noted that we would incorporate the suggestion into the paper to improve clarity. We hope that these few additions help clarify any concerns.

We specifically would like to thank reviewer 73yj for appreciating the strength of our proof and for indicating that our method is highly suitable for CCBs. We thank reviewer aW5H for re-iterating that the problem has several real-life applications, for example in online advertising. Further, we thank reviewer oUko for recognising the novelty of our formulation of the causal contextual bandit problem as a convex minimization problem, and for noting that our bounds are tight. We would like to thank reviewer fZFX for appreciating the clarity of our proposed algorithm, and for highlighting that our problem setting is new.

We thank all the reviewers for spending their valuable time to review our manuscript.


Thank you!

---

### Meta-Review · Area_Chair_fjMT · 2023-12-04

**Metareview:**

The reviewers all had some (common) concerns about the acute sensitivity of the performance to $\lambda$, comparison to other state-of-the-art exploration strategies, comparison to causal bandit works and the limited empirical evaluations. It was somewhat unfortunate that there were limited interactions between the authors and the reviewers but the AC believes that a rewrite would be necessary for another round of evaluation.

**Justification For Why Not Higher Score:**

There were several limitations of the work as discussed above.

**Justification For Why Not Lower Score:**

The score is, at this point in time, rather lukewarm.

---

### Decision · Program_Chairs · 2024-01-16

Reject